# Tumor Immune Microenvironment and Immunosuppressive Therapy in Hepatocellular Carcinoma: A Review

**DOI:** 10.3390/ijms22115801

**Published:** 2021-05-28

**Authors:** Kyoko Oura, Asahiro Morishita, Joji Tani, Tsutomu Masaki

**Affiliations:** Department of Gastroenterology and Neurology, Faculty of Medicine, Kagawa University, 1750-1 Ikenobe, Miki, Kita, Kagawa 761-0793, Japan; asahiro@med.kagawa-u.ac.jp (A.M.); georget@med.kagawa-u.ac.jp (J.T.); tmasaki@med.kagawa-u.ac.jp (T.M.)

**Keywords:** hepatocellular carcinoma, tumor microenvironment, immunotherapy, immune checkpoint inhibitor, molecular target agent, adaptive cell transfer-based therapy, cytokine-induced killer, chimeric antigen receptor, chronic hepatitis, fibrosis

## Abstract

Liver cancer has the fourth highest mortality rate of all cancers worldwide, with hepatocellular carcinoma (HCC) being the most prevalent subtype. Despite great advances in systemic therapy, such as molecular-targeted agents, HCC has one of the worst prognoses due to drug resistance and frequent recurrence and metastasis. Recently, new therapeutic strategies such as cancer immunosuppressive therapy have prolonged patients’ lives, and the combination of an immune checkpoint inhibitor (ICI) and VEGF inhibitor is now positioned as the first-line therapy for advanced HCC. Since the efficacy of ICIs depends on the tumor immune microenvironment, it is necessary to elucidate the immune environment of HCC to select appropriate ICIs. In this review, we summarize the findings on the immune microenvironment and immunosuppressive approaches focused on monoclonal antibodies against cytotoxic T lymphocyte-associated protein 4 and programmed cell death protein 1 for HCC. We also describe ongoing treatment modalities, including adoptive cell transfer-based therapies and future areas of exploration based on recent literature. The results of pre-clinical studies using immunological classification and animal models will contribute to the development of biomarkers that predict the efficacy of immunosuppressive therapy and aid in the selection of appropriate strategies for HCC treatment.

## 1. Introduction

### 1.1. Hepatocellular Carcinoma (HCC)

The incidence of liver cancer has increased worldwide, not only in East Asia but also in western Europe and the United States. Liver cancer currently ranks sixth in incidence rate and fourth in mortality rate of all cancers [1]. Hepatocellular carcinoma (HCC) is the most common subtype, accounting for more than 90% of all primary liver cancers [2]. Multiple etiological risk factors are associated with the incidence of HCC, including chronic infection with hepatitis B virus (HBV) and/or hepatitis C virus (HCV), alcohol abuse, non-alcoholic steatohepatitis (NASH), autoimmune liver disease, drug-induced liver injury, and aflatoxin exposure [2,3]. Despite great therapeutic advances, HCC has one of the worst prognoses with a 5-year survival rate of 15–38% in the United States [4,5] and Asia [6] due to late diagnosis, resistance to chemotherapy, and frequent recurrence and metastasis.

Treatment options such as surgical resection, radiofrequency ablation, and transarterial chemoembolization are effective for HCC localized in the liver, while systemic therapy with various drugs targeting the tumor microenvironment (TME) is available for unresectable HCC. Since sorafenib was first shown to prolong the survival of patients with unresectable HCC [7], systemic therapy with molecular-targeted agents (MTAs) has continued to evolve significantly as a useful therapeutic strategy for advanced HCC. Multikinase inhibitors such as sorafenib, lenvatinib, regorafenib, and cabozantinib, as well as the vascular endothelial growth factor (VEGF) inhibitor ramucirumab, have found widespread clinical applications [8,9,10,11]. In addition to MTAs, new therapeutic strategies such as cancer immunosuppressive therapy based on immune checkpoint inhibitors (ICIs) have progressed in recent years. For advanced HCC, the combination of ICIs and VEGF inhibitor has shown better results than sorafenib [12], and the combination of atezolizumab and bevacizumab (atezo+bev) is now positioned as the first-line therapy for patients with advanced HCC. Although systemic therapies for HCC have undergone a major paradigm shift, treatment for advanced HCC remains inadequate because of a lack of evidence associated with treatment resistance and prediction of treatment response. Since the efficacy of immunosuppressive therapy including ICIs depends on the tumor immune microenvironment, it is necessary to elucidate the immune environment of HCC to improving current core treatment strategies and prognosis for HCC patients.

### 1.2. Tumor Microenvironment (TME)

In addition to cancer cells, the TME includes innate and adaptive immune cells, stromal cells, endothelial cells, and cancer-associated fibroblasts. In the TME, immune cells such as macrophages infiltrate, fibroblasts proliferate, and angiogenesis is induced, and the TME is reportedly deeply associated with the formation, survival, and metastasis of tumor tissues [13]. Furthermore, the development of treatment approaches that target angiogenesis, adhesion, and infiltration of tumor cells in the TME is also in progress. Although previous studies have focused on adaptive immune cells due to the potent cytotoxicity of T lymphocytes in the context of cancer, current TME-targeted treatments have predominantly focused on the innate T cell immune responses, including checkpoint blockade and chimeric antigen receptor (CAR) T cell therapies. In the treatment of advanced HCC as well as other cancer types, immunotherapeutic approaches have increasingly focused on monoclonal antibodies against cytotoxic T lymphocyte-associated protein 4 (CTLA-4) and programmed cell death protein 1 (PD-1), which lock the immune checkpoint inhibition pathways [14]. Therefore, exploring the immune microenvironment associated with these treatment strategies contributes to improved outcomes in patients with advanced HCC.

In this review, we summarize the immune TME of HCC and immunosuppressive approaches focused on monoclonal antibodies against CTLA-4 and PD-1 and ongoing treatment modalities based on recent literature. Although clinical trials validating the efficacy of immunosuppressive therapies are to proceeding at present, we also describe the results of pre-clinical studies using immunological classification and animal models, which will contribute to the development of biomarkers that improve the efficacy of treatment and patient prognosis.

## 2. The Constituent Cells and the Immune Microenvironment in the Liver

The liver is an immune organ that is rich in immunocompetent cells, including Kupffer cells (KCs), liver sinusoidal endothelial cells (LSECs), hepatic stellate cells (HSCs), pit cells, and lymphocytes such as natural killer (NK) T cells, gamma delta (γδ) T cells, and dendritic cells (DCs) [15]. The factors contributing to hepatic immunity include the origins of the liver as a hematopoietic organ during the embryonic period, the flow of the portal vein blood from the gastrointestinal tract and spleen into the liver, and the formation of mucosal immunity via excretion of metabolites from the biliary system. LSECs do not have a basement membrane, unlike general capillary endothelial cells, to facilitate substance exchange between hepatocytes and blood, and lymphocytes come into direct contact with hepatocytes. As the liver is constantly exposed to bacterial components and dietary antigens that flow from the gastrointestinal tract via the portal vein, it is necessary to maintain a level of tolerance to balance bacterial pathogen elimination and avoid excessive inflammation induced by the non-pathogenic intestinal environment. Furthermore, the liver produces immune-related molecules such as the C-reactive protein (CRP) and soluble pattern recognition receptors, which play pivotal roles in systemic inflammation and immunity [15].

### 2.1. Kupffer Cells (KCs)

KCs are tissue macrophages that occur in the lumen of the liver sinusoids, adhere to sinusoidal endothelial cells with their protrusions, and maintain tolerance to gastrointestinal antigens, forming the first line of defense against pathogens [16]. In a normal liver, this tolerogenic phenotype of KCs is important to prevent excessive immune responses to invading immunoreactive materials, including gastrointestinal-derived materials and antigens present on dead or dying cells [17]. KCs are activated by endotoxins, complements, and other pathogen-associated molecular patterns through toll-like receptor (TLR) expression. Various cytokines and chemokines produced by KCs subsequently activate neutrophils, which suppress the inflammatory response by ingesting bacteria and carrying out apoptosis after clearance by KC [18]. Furthermore, KCs are the first line of defense against cancer cells that metastasize from multiple organs and pathogens. The antimetastatic effects of KCs include phagocytosis of tumor cells, apoptosis during the early invasion process, production of the tumor necrosis factor (TNF)-α, and mobilization of neutrophils and NK cells [19].

### 2.2. Liver Endothelial Cells (LSECs)

LSECs occur mainly in the sinusoidal wall of the liver and have minimal basement membranes in the endothelium and open fenestrations that facilitate the steric transport of cargo from the sinusoidal space to the parenchyma and the space of Disse [20,21]. LSECs are involved in the regulation of sinusoidal blood flow through carbon monoxide production. When the function of KCs deteriorates, LSECs uptake and process foreign substances in the blood. LSECs are clusters of differentiation 4 (CD4) +, express major histocompatibility complex (MHC) class II proteins, undergo excessive endocytosis, and have scavenger receptors that support the most potent scavenger system in the body [22,23]. Antigen uptake and antigen presentation via scavenger receptors of LSECs are key steps in promoting T cell tolerance under physiological conditions. Furthermore, LSECs play important roles in immune regulation within the liver through leukocyte adhesion, bacterial processing, and viral clearance [21,24]. LSECs also contribute to leukocyte trafficking by recruiting these leucocytes from the peripheral circulation into the liver. Through a multi-step adhesion cascade that involves several receptor-ligand interactions, the activated endothelium captures insulating immune cells and allows leukocyte migration to sites of tissue injury or infection [24]. Thus, in the liver, lymphocytes frequently migrate via the transcellular route, whereas in other organs they migrate through the paracellular route between endothelial cell junctions [25]. As excessive immunosuppressive leukocyte subsets promote pathogen escape and tumorigenesis, LSECs also play active roles in regulating the microenvironment in persistent hepatitis.

### 2.3. Hepatic Stellate Cells (HSCs)

HSCs are lipid-storing cells in the space between hepatocytes and LSECs on the hepatic lobule and contain 50–80% of all vitamin A in the body [26]. By stimulating signaling molecules such as transforming growth factor (TGF)-β1, mainly released from KCs, HSCs produce collagen fibers and extracellular matrix constituents and are directly involved in liver fibrosis [27]. Activated HSCs produce leptin, which enhances liver fibrosis through the upregulation of TGF-β expression and the mechanism of autocrine activation in HSCs [28]. However, adiponectin inhibits HSC activation and suppresses liver fibrosis. HSCs are rarely involved in the direct immune response.

### 2.4. Pit Cells (Natural Killer Cells)

Pit cells are components of the innate immune system and liver-specific NK cells, which constitute the most important first line of defense against invasion of cancer cells by mediating cytotoxicity and cytokine production; NK activity of the liver is the strongest compared to that in other organs [29,30]. NK cell function is strictly regulated by the balance of active and inhibitory receptors that bind to specific ligands and interact with target cells. MHC-1 is expressed on normal hepatocytes and suppresses pit cell activation by interacting with inhibitory receptors with these cells. Pit cells can directly stamp out infected cells that lack MHC-1 expression [31]. However, pit cells lose the inhibitory signal by interacting with the NK inhibitory receptor with the MHC-1 complex and can be activated to kill infected hepatocytes via downregulation of MHC-1 by a viral infection or tumorigenesis [32]. Pit cells appear to play a pivotal role in the suppression of HCC progression. Therefore, a method that utilizes the NK activity of pit cells is expected to be an effective therapeutic strategy for HCC [32].

### 2.5. Hepatic Lymphocytes

In the liver, lymphocytes reside throughout the parenchyma and the portal tracts, including innate (NKT cells) and adaptive immune systems. Conventional T cells comprise CD8+ and CD4+ T cells, and the liver has a high proportion of CD8+ T cells; more T cells differentiate outside the thymus than in the peripheral blood [33]. In the liver, the proportion of effector and memory T cells, which play a central role in the defense against invasion by pathogens and cancer cells, is also higher than that in peripheral blood.

NKT cells account for 20% of all intrahepatic lymphocytes and have a unique subset of mature CD4+ T cells that can simultaneously express both NK cell-surface proteins and T cell receptors (TCRs) [33,34]. Hepatocytes express large amounts of CD1d, a member of the major MHC class I-like molecule family, which specifically contains lipid antigens [35]. CD1d mainly promotes the differentiation and activity of NKT cells and affects NKT cell activity throughout the body. CD1d-restricted NTK cells are classified into two subtypes (types I and II) according to antigen specificity and TCR diversity. Type I NTK cells recognize self-lipids and microbial lipid antigens, and α-galactosyl ceramide (GalCer) is the most effective ligand for their activation [36]. Activated type I NKT cells produce various inflammatory cytokines that subsequently stimulate pit cells, lymphocytes, and DCs [34]. In contrast, type II NKT cells express various TCRs that activate self-lipids, cannot express α-GalCer [35,36] and are more likely to produce Th1 cytokines, which inhibit the activation of proinflammatory type I NKT cells and protect against liver injury [34,36].

γδT cells rapidly respond to microorganisms and stress proteins to produce interferon (IFN)-γ and protect against infection. Before migration to peripheral tissues, γδT cells are produced in the thymus, but they are enriched in the liver and exhibit liver-specific features in liver infection or inflammatory liver diseases [37]. Most T cells have TCRs composed of two glycoproteins called α-chain and β-chain, whereas γδT cells have TCRs consisting of a γ-chain and a δ-chain. γδT cells possess oligoclonal or invariant TCRs that recognize stress proteins or non-protein antigens [33].

### 2.6. Dendritic Cells (DCs)

DCs are antigen-presenting cells capable of activating naïve T cells. There are various DCs such as progenitor, immature, and mature cells in the liver, but most of them are predominantly immature cells in the healthy liver [38]. When DCs take up antigens, they move to the regional lymph node and mature to present the antigen to unsensitized T cells. In the liver, there are myeloid DCs that act mainly on the Th1 system and a subset that mainly acts on the Th2 system. Resident hepatic DCs are derived from the bone marrow and are located around the central and portal veins [33]. DCs play an important role in tumorigenesis and viral infection [39].

## 3. The Immune Microenvironment in Liver Inflammation and Fibrosis

During chronic liver injury caused by pathogens such as the hepatitis virus and parasites or by drugs, alcohol, and steatosis, both the systemic and local immune microenvironments regulate liver inflammation and fibrosis, leading to hepatocarcinogenesis. In addition, liver cirrhosis is also a state of immune dysfunction caused by the excessive activation of proinflammatory cytokines [40]. In patients with liver cirrhosis, the risk of bacterial infection and sepsis increases due to the abnormal gastrointestinal barrier permeability and bacterial translocation; therefore, it is useful to elucidate the immune microenvironment during chronic liver disease to improve prognosis [41].

### 3.1. Immune Regulation and Microenvironment in Liver Inflammation

Inflammatory mediators produced by liver-resident immune cells play pivotal roles in maintaining local liver and systemic homeostasis, and resident myeloid cells contribute to the maintenance of hepatic tolerance [42]. In response to bacterial endotoxins, KCs produce anti-inflammatory cytokines such as interleukin (IL)-10 and prostaglandins, which subsequently suppress the expression of co-stimulatory molecules on antigen-presenting cells, preventing CD4 + T cell activation. Compared with those in the spleen, hepatic myeloid DCs also produce significantly more IL-10 and suppress T cell activation [42,43]. In addition, myeloid-derived suppressor cells (MDSCs) produce immunosuppressive cytokines, including IL-10 and TGF-β, to maintain a tolerogenic liver environment.

The liver also plays a central role in the detection of and response to inflammatory signals. Cytokines produced by extra-hepatic immune cells are detected by hepatocytes in the hepatic blood flow following an increase in acute-phase protein production and synthesis of IL-6, which increases the acute systemic response [42]. Acute-phase proteins produced directly by hepatocytes promote systemic inflammatory responses, such as massive immune cell infiltration to the initial inflammatory site, while controlling the processes necessary to suppress excessive inflammation. These processes include suppression of neutrophil function by protease inhibitors, decreased TNF production, suppressive MDSC recruitment by amyloid A, control of bystander tissue damage by the inflammatory process, and enhancement of the repair process [44].

Liver-resident immune cell populations and inflammatory mediators such as IL-1α, TNF-α, and IL-6 are indispensable for liver regeneration after liver injury and control metabolism and xenobiotic detoxification. Reports demonstrating the importance of these inflammatory mediators in liver regeneration show that targeted IL-6 disruption caused hepatic regeneration disorder in mice. Its function recovered with a single dose of IL-6, and impaired liver generation was also observed in mice treated with antibodies targeting TNF-α [42].

### 3.2. Hepatic Immune Microenvironment with Progression to Fibrosis

Liver fibrosis is the final destination of pathological hepatic disorders in chronic liver diseases such as viral hepatitis, alcoholic liver disease, and liver steatosis. HSCs and KCs play important roles in the progression of liver inflammation to fibrosis. Several proinflammatory factors, including TGF-β and platelet-derived growth factor (PDGF), activate HSCs through TLR4, which then produce extracellular matrix proteins such as various subtypes of collagen [45]. TLR4 has been shown to play important roles in regulating liver damage, and mice with hepatocyte-deleted TLR4 were reportedly protected against chronic alcoholic liver disease and fatty liver [46]. In addition to TLR4, HSCs also express TLR3 and TLR9, which are related to liver fibrosis. The TLR3 ligand, polyinosinic-polycytidylic acid, activates NK cells through high expression of TNF-α-related apoptosis-inducing ligand (TRAIL) and induces cell death of activated HSCs, thus resulting in reduced severity of liver fibrosis [47]. TLR9 upregulates HSCs under the influence of host origin DNA from apoptotic DNA, enhancing liver fibrosis [48]. KCs also cause liver fibrosis through activation of TLRs by lipopolysaccharides, triggering the production of several cytokines, including TGF-β. Among other pattern recognition receptors, besides TLRs, only the NOD-like receptor (NLR) is associated with liver fibrosis progression. NLRs form bioactive protein complexes called inflammasomes, which produce proinflammatory cytokines such as IL-1β and IL-18, resulting in the activation of HSCs in chronic inflammatory liver diseases [49]. Recently, selective inflammasome inhibitors have been shown to exert proactive effects in cholestatic liver injury and liver fibrosis in a mouse model [50].

NK cells inhibit the progression of liver fibrosis by killing activated HSCs [51]; therefore, the association between NK cells and fibrosis in various liver diseases has been highlighted. In HCV-infected patients, the accumulation of NK cells with highly expressed NKp64 receptors, which showed potent cytotoxic activity and IFN-γ secretion, was inversely correlated to HCV-RNA levels and the degree of liver fibrosis [52]. In a mouse model of chronic alcohol consumption, induction of TSC resistance to NK cell killing, desensitization of HSC resistance from NK toxicity, and inhibition of IFN-γ accelerated liver fibrosis [53]. Signal transduction and activation of transcription 1 (STAT1) signaling antagonize the effects of TGF produced by HSCs and negatively regulate fibrosis to support NK cytotoxicity [16].

NKT cells, which are mainly present in the liver, are also central immune players in the progression from liver inflammation to fibrosis; however, their function in fibrosis seems to be uncharacterized and influenced by various conditions [54]. In mice lacking mature NKT cells caused by disruption of the CD1d molecule, thioacetamide-induced hepatocellular inflammation and damage were ameliorated, and the profibrogenic response of tissue inhibitor of matrix metalloproteinase (TIMP) 1 was significantly reduced [55]. In a study of a fibrosis-induced model with carbon tetrachloride (CCl4), NKT cell-deficient mice were found to be more susceptible to CCl4-induced liver inflammation. Although strong activation of NKT cells by α-galactosyl ceramide accelerates CCl4-induced liver fibrosis, CCl4 administration induced only a slightly higher degree of liver fibrosis in NKT cell-deficient mice than that in control mice at 2 weeks but not 4 weeks after induction by CCl4. During chronic liver injury, NKT cells inhibit liver fibrosis in the early stage but may not affect the late stage due to the depletion of NKT cells [56].

## 4. Immune Network in HCC Microenvironment

### 4.1. Cancer-Immunity Cycles

The TME in HCC comprises cancer cells, immune cell subsets, the intricate cytokine environment, and the extracellular matrix and plays a major role in tumor progression [57]. Such a complex and dynamic intertumoral immune system is described by the following seven-step immunity cycle (Figure 1) [58]. Cancer antigens are released from cell-dead cancer cells (Step 1). DCs take up these cancer antigens and present them in the lymph nodes (Step 2). T cells are primed and activated (Step 3). T cells migrate to tumor tissue (Step 4). Activated T cells infiltrate the tumor (Step 5). T cells recognize (Step 6) and damage cancer cells (Step 7). Similar to the therapeutic strategies in many other cancer types, one or more of these steps are impaired in HCC to elucidate an effective cancer immune response and to contribute toward tumor resistance and progression.

As an activator of the cancer-immunity cycle, high mobility group box 1 is released simultaneously with cancer antigens to act on TLR2 and TLR4 of DCs to promote DC maturation and induce antitumor immune responses when cancer cells cause immunogenic cell death [59]. In addition, C-C chemokine receptor type 7 (CCR7) induces DC migration to lymph nodes. Stimulation of TRL activates DCs, and DC maturation increases the expression level of co-stimulatory molecules such as DC40, resulting in DCs having a high antigen-presenting ability. It has been reported that a good prognosis is associated with high amounts of conventional DC 1 (cDC1), which take up the cancer antigens in the tumor, migrate to the lymph nodes, and cross-prime CD8+ T cells [59,60,61]. It has also been reported that NK cells produce X-C motif chemokine ligand 1 (XCL1), C-C motif ligand 5 (CCL5), and fms-like tyrosine kinase 3 (FLT3) to attract cDC1 [60,61]. However, chemokines such as the C-X-C motif ligand 9 (CXCL9), CXCL10, and CCL5 present in the intertumoral environment strongly induce T cell migration to tumors. The interaction between T cell-expressed lymphocyte function-associated antigen 1 (LFA1) and intercellular adhesion molecule 1 (ICAM1) on the intravascular endothelium activates T cell infiltration into the tumor tissue. IFN-γ produced by T cells and NK cells suppresses cancer cell proliferation and enhances the MHC class 1 antigen presentation pathway. As a suppressor of the cancer-immunity cycle, lymph node regulatory T cells interact with CTLA-4 and co-stimulatory molecules such as CD80 and CD86 expressed on DCs to suppress T cell priming. To suppress T cell infiltration into tumor tissues, cancer cells produce VEGF, phosphatase, and tensin homolog deleted on chromosome 10 (PTEN), which activates the phosphatidylinositol 3 kinase (PI3K)/AKT pathway [62]. Alternatively, cancer cells with activated β-catenin signals suppress the accumulation of cDC1 in cancer cells, resulting in decreased production of CXCL10 [63]. The deficiency of transporters associated with antigen processing (TAP) or β2 macroglobulin (B2M) involved in MHC class 1 antigen presentation in cancer cells causes them to escape T cell recognition. In step 7 of the immunity cycle, wherein T cells damage cancer cells, immune checkpoint molecules such as PD-1-expressing T cells and PD-ligand 1 (PD-L1)-expressing cancer cells act as co-inhibitors that suppress or halt T cell immune responses [64].

Furthermore, subsets of immune cells such as MDSCs, tumor-associated macrophages (TAMs), tumor-associated neutrophils (TANs), cancer-associated fibroblasts (CAFs), regulatory T cells (Tregs), tumor-infiltrating lymphocytes (TILs), and CD8+ cytotoxic T lymphocytes (CTLs) are also critical immunosuppressive components in the TME of HCC (Table 1).

### 4.2. Myeloid-Derived Suppressor Cells (MDSCs)

MDSCs are a heterogeneous population of immature myeloid cells that originate in the bone marrow, appear in the local tissue and systemic blood due to inflammation or cancer, and play a major role in suppressing antitumor immunity in tumor-bearing hosts [86,87]. Several tumor-associated cytokines, including granulocyte-macrophage colony-stimulating factor (GM-CSF), granulocyte-colony-stimulating factor (G-CSF), IL-β, IL-6, VEGF, and monocyte chemotactic protein (MCP)-1 reportedly caused MDSC accumulation and regulated their migration in a mouse model of HCC [65]. In another recent report, it was shown that a cell cycle-related kinase unique to HCC upregulates IL-6 production and activates nuclear factor-κB (NF-κB) by enhancing the zeste homolog 2 (EZH2) enzyme, resulting in the accumulation of polymorphonuclear MDSCs with potent T cell suppression in the TME [66]. HCC-related CAFs can attract monocytes via the CXCR4 pathway, which is a receptor for stromal cell-derived factor (SDF)-1a, and induce their differentiation into MDSCs by activating IL-6-mediated STAT3 [88]. Hypoxemia is an important factor in the TME in solid tumors, including HCC, and MDSCs have been demonstrated to preferentially invade the hypoxic region of human HCC tissue through the CCL26/C-X3-C motif chemokine receptor 1 (CX3CR1) pathway [67]. In addition, hypoxia-inducible factor 1α (HIF-1α) promotes MDSC accumulation by converting extracellular ATP to 5′ AMP through ectonucleoside triphosphate diphosphohydrolase 2 (ENTPD2) overexpression in HCC cells [68].

In HCC, MDSCs exert powerful immunosuppressive effects by expanding immune checkpoint signaling and reducing NK cell cytotoxicity. MDSCs express galectin-9, a ligand for T cell immunoglobulin and mucin domain 3 (TIM-3), a surface antigen on T cells, and induces apoptosis [89]. In advanced HCC patients, MDSCs induce PD-L1 expression by interacting with KCs, and the inhibition of autologous NK cell cytotoxicity and cytokine release is mainly dependent on the NKp30 receptor [90]. In addition, MDSCs have been reported to be potential therapeutic targets for resetting the immunotolerant state of HCC. Combination immunotherapy with anti-PD-1/PD-L1 and concomitant targeting of MDSCs such as p38 mitogen-activated protein kinase inhibitors may enhance the efficacy of eradicating tumorigenicity and may prolong survival in the fibrotic HCC mouse model [91]. Another recent study suggested that combination therapy with radiation and IL-12 significantly elevates antitumor immunity in HCC tissues by reducing the accumulation of tumor-infiltrating MDSCs and reactive oxygen species production [92].

### 4.3. Tumor-Associated Macrophages (TAMs)

TAMs that infiltrate and accumulate in the tumor are in an unusually activated state and include tissue-resident macrophages and monocyte-derived exudate macrophages mobilized by inflammatory stimuli. The proportion of these TAMs varies depending on the cancer type; however, exudate TAMs become the main component as the tumor size increases [93] because MCP-1, GM-CSF, and macrophage (M)-CSF produced by cancer cells and TAMs induce the migration of monocytes to the tumor stroma and their differentiation into TAMs [94]. In another classification based on the state of macrophage activation, TAMs also consist of two groups, namely those that undergo classical activation (M1) by IFN-α/β or IFN-γ, which is the TLR ligand and Th1 response index, and those that undergo alternative activation (M2) by Th2 type or other anti-inflammatory factors such as IL-4 and IL-10. The M1/M2 balance also differs depending on the cancer type. M1-like cells produce antitumor factors such as TNF-α and nitric oxide and induce antitumor immune responses through their T cell-stimulating activity. In contrast, M2-like cells not only produce tumor growth factors such as IL-6 but also produce angiogenic molecules, including VEGF and immunosuppressive factors such as arginase 1 (Arg1), IL-10, TGF-β, and indoleamine 2,3-dioxygenase (IDO), resulting in tumor promotion [95].

Research into understanding the role of immunosuppressive TAMs in HCC has recently attracted attention. It has been demonstrated that TAMs can produce various chemokines, including CCL17, CCL18, and CCL22, resulting in the attraction of Treg cells to tumor tissues and activation of CTLs [69,70]. CCL2 is a highly expressed cytokine in HCC and plays a vital role in the migration of inflammatory monocytes, circulating precursors of tissue macrophages [96]. The CCL2/CCR2 axis promotes the differentiation of CCR2+ inflammatory monocytes into TAMs and is a potential immunotherapeutic target [97,98]. It has recently been reported that the necrotic debris of HCC cells induces potent IL-1β release by TAMs via the TLR4/TIR domain-containing adapter-inducing interferon (TRIF)/NF-κB pathway under persistent hypoxia, thus promoting EMT and HCC immune evasion [71]. Several proinflammatory cytokines, including TNF-α, IL-β, IL-6, and IL-23, are produced by TAMs in the tumor stroma of HCC and induce the expansion of IL-17-producing CD4+ Th17 cells that suppress antitumor immunity by overexpressing PD-1, CTLA-4, and glucocorticoid-induced TNF receptor family-regulated protein (GITR) [72]. In addition, TGF-β derived from the TME in HCC promotes TIM-3 expression in TAMs. Activated TAMs promote tumor growth and immune tolerance via the NF-κB/IL-6 pathway [73].

TAMs are associated with prognosis in patients with HCC. The low presence of CD86+ TAMs and high presence of CD206+ TAMs in HCC tissues are significantly associated with an advanced clinical stage, poor overall survival (OS), and increased time to recurrence (TTR) [99]. In addition, inhibiting TAMs may be a potential strategy for future treatment of HCC. It has been demonstrated that cell-derived Wnt ligands stimulate M2-like cells to induce TAM polarization via the Wnt/β-catenin pathway, whereas blocking Wnt signaling in cancer cells or Wnt/β-catenin pathway activation in TAMs contributes to controlling HCC progression [100].

### 4.4. Tumor-Associated Neutrophils (TANs)

TANs play major roles in linking inflammation and cancer development and are significantly associated with cancer progression and metastasis [101]. Although several bone marrow-derived cells promote tumor progression [102], TANs affect tumor biological behaviors differently depending on the polarization, for example, whether macrophages have the antitumor or protumor phenotype [103]. The plasticity of these subtypes depends on the presence of TGF-β [104]; TGF-β with increased inhibition of IFN-β occurs in the N1 phenotype, whereas neutrophils polarize TGF-β in the N2 phenotype [105]. Type I IFNs modulate TAN activation and alter the neutrophilic antitumor phenotype in both mice and humans. TANs also mainly suppress antitumor immunity by inducing apoptosis of CD8+ T cells through nitric oxide production-mediated TNF-α [75].

In TMEs, TANs are associated with cancer development as they promote cellular transformation, antitumor immunity, and tumor progression [106,107]. In HCC, the presence of intertumoral neutrophil infiltration was reported to indicate a poor prognosis associated with the overexpression of CXCL5 [74]. Cytokines, CCL2, and CCL17were expressed most highly by TANs and HCC cell-activated peripheral blood neutrophils. These protein levels were significantly higher in TANs than in the peripheral blood, and the number of CCL2+ or CCL17+ TANs was associated with tumor size, microvascular invasion, differentiation, and clinical stage. In addition, it was shown that TAN-conditioned media increased the migratory activity of macrophages and Tregs and recruited them into HCC tissue to promote tumor progression and resistance to sorafenib [106].

A recent report indicated that the loss of hypoxia-associated factor (HAF) encoded by spliceosome-associated factor 1 (SART1) induces an increase in regulated upon activation normal T cell expressed and secreted (RANTES) levels, which are HIF-1α-dependent chemokines [108]. The activation of the HIF-1α/RANTES-driven pathway results in TAN infiltration associated with HCC initiation and progression in NASH. As a positive feedback loop that regulates cancer stem-like cells and TANs in HCC, upregulation of microRNA (miR)-301b-3p causes hyper-action in NF-κB signaling and high CXCL5 secretion, resulting in TAN infiltration [109]. Thus, TANs strongly contribute to the suppression of TMEs, but the direct interaction between TANs and HCC cells is not yet fully understood.

### 4.5. Cancer-Associated Fibroblasts (CAFs)

Fibroblasts are important cells that form the extracellular matrix, such as collagen fibers that support various tissues, and CAFs promote carcinogenesis of normal epithelial cells in the TME and endow cancer cells with properties similar to those of stromal cells. CAFs differentiate from cancer cells and vascular cells or are derived from activated mesenchymal stem cells present in the bone marrow and settle in the blood circulation.

Recent evidence indicates that CAFs induce MDSC generation via the IL6/STAT3 axis and SDF-1α [88]. As an indirect effect of the TME, CAF-derived IDO and prostaglandin E2 (PGE2) attenuate TNF-α and IFN-γ production by NK cells and are associated with HCC development [76]. Bone morphogenetic protein 4 (BNP-4) may be one of the most important regulators of CAF function in HCC and activates hepatic fibroblasts to secrete cytokines, enhancing their invasiveness [77]. As a new therapeutic target, the important roles of CAFs in the TME may contribute to the suppression of liver fibrosis and HCC progression [110,111,112,113].

### 4.6. Regulatory T Cells (Tregs)

Tregs are a subset of CD4+ T cells identified with the CD25 marker and mainly expressed by forkhead box P3 (FOXP3), a transcription factor; they also play an important role in maintaining autoimmune tolerance and homeostasis. Therefore, the primary role of Tregs is to suppress the excessive immune response that causes autoimmune, inflammatory, and allergic diseases. However, excessive work in the TME helps tumor progression. Many effector Tregs with strong inhibitory activity infiltrate the local tumor area in various cancer types, such as melanoma, and suppress the antitumor immune response [114]. Tregs are mobilized by regulating the CCL6/CCL20 axis and are activated by TCR engagement with IL-10 and TGF-β signaling [115].

It has been reported that the number of Tregs increases in the tumor tissue or peripheral blood of HCC patients relative to that in healthy individuals. Thus, the proportion and the absolute number of CD4+ CD25+ T cells significantly increase in the area surrounding the tumor [116], and Tregs increased in the peripheral blood of HCC patients compared to healthy subjects [78]. Long noncoding RNAs (lncRNAs), as well as proinflammatory signals, may play important roles in promoting Treg differentiation and HCC progression. Overexpression of lnc-epidermal growth factor receptor (lncEGFR) in Tregs prevents its ubiquitination by activating downstream the activator protein 1 (AP-1)/nuclear factors of activated T cells 1 (NFAT1) axis in Tregs, which results in the promotion of immunosuppression in HCC. In addition, Tregs may be a significant target for the immune suppression of HCC, and sorafenib, a key multikinase inhibitor for HCC treatment, reduces the frequency of Treg infiltration into the liver by suppressing the TGF-β signaling [79].

### 4.7. Tumor-Infiltrating Lymphocytes (TILs)

TILs consist of T cells, B cells, and NK cells, which are typical components of the host’s antitumor immune response. The surface of TILs contains several antigens, including CD3, CD4, CD8, CD16, CD20, CD56, CD57, CD68, and CD169 [117]. FOXP3 is the master transcription factor of *Tregs*, and FOXP3+ T cells are representative TILs that function as tumor suppressors. Many immunohistochemical studies targeting these specific antigens have indicated that specific types of immune cells not only regulate the host’s antitumor immune environment but also create conditions that promote tumor growth in the TME [80].

Through the intrahepatic immune response, diverse lymphocyte populations contribute differently to HCC immune surveillance. To evaluate whether HCC is tumor-promoting or inhibitory, it is important to evaluate the intertumoral balance of regulatory and cytotoxic T cells [118]. TILs may be useful biomarkers associated with the prognosis of patients with HCC. Although there is a report that high filtration of FOXP3+ T cells is associated with poor prognosis [81], others have reported no correlation between cell infiltration and OS [119]. A high density of CD8+ T cells has been shown to have a poor prognosis [120], but another report indicated the opposite result [121]. The impact of TILs on the prognosis of HCC is controversial, but a recent meta-analysis has shown that the density of FOXP3+, CD8+, CD3+, and granzyme B+ lymphocytes in TILs is significantly associated with improved survival, and these subsets can serve as prognostic biomarkers in HCC [117].

### 4.8. CD8+ Cytotoxic T Lymphocytes (CTLs)

CTLs are a type of T cell that express CD8 on the cell membrane and recognize abnormal cells such as cancer cells by receiving instructions from helper T cells. CTLs have cytotoxic activity only after activation. The TCRs on naïve CD8+ T cells that do not have cytotoxic activity recognize foreign antigens together with MHC class I antigen-presenting cells. When a signal from co-stimulating molecules is input, naïve CD8+ T cells become CTLs with specific cytotoxic activity against cells presenting foreign antigens. CTLs release perforin, granzyme, and TNF to destroy cancer cells. Most of these CTLs die once their work is done, but some remain as memory killer cells to recognize the same cancer cells in the future.

Although the antitumor immune response is strictly suppressed by various mechanisms, the presence of CD8+ CTLs in HCC contributes to improved survival. Hypoxia, metabolic competition with HCC cells, lack of CD4+ T cells, and high expression of several regulatory molecules, including VEGF, CXCL17, IL-10, and IDO, cause restriction of tumor-associated antigen (TAA)-specific response and poor production of IFN-γ by CTLs [67,82,83,84]. In addition, Expression of Fas and its ligand expression in CD8+ T cells is associated with antitumor immunity in HCC [122]. Fas ligand expression in endothelial cells is induced by emphasizing VEGF-A and PGE2, which results in the excessive turnover of CD8+ cells and reduction of antitumor response [83]. CTLs are also suppressed by IL-2 and IDO production by CD14+ DCs [85].

## 5. Current HCC Therapeutic Strategies Targeting Multikinase Activity and the Immune System

The complex interaction of immune cells with effector molecules in the TME around HCC alters the immune system and promotes or suppresses HCC growth. Elucidating the multimodal mechanisms of these immunotherapies might help to improve early and advanced HCC outcomes.

Systemic therapies targeting typical tumor-related pathways in the TME, including VEGF-dependent angiogenesis, PI3K/AKT/mammalian target of rapamycin (mTOR), adenosine 5′-monophosphate-activated protein kinase (AMPK), and c-mesenchymal-epithelial transition factor (c-MET), are approved or under clinical trials for several cancer types, but evidence of their therapeutic efficacy in HCC is limited [123]. However, molecular-targeted therapies for advanced HCC have evolved significantly in recent years. Sorafenib exhibits antitumor effects by inhibiting vascular endothelial growth factor receptor (VEGFR), Raf-1, B-Raf, and platelet-derived growth factor receptor (PDGFR). It was for a long time the first MTA that prolonged the survival of patients with advanced HCC and the only systemic chemotherapeutic drug that was indicated for HCC [7]. Lenvatinib is a multikinase inhibitor that targets VEGFR1-3, FGFR1-4, PDGFRα, and tyrosine kinase receptors, thereby suppressing neo-vessel assembly and maturation and decreasing vascular permeability and TMEs [124]. In the REFLECT trial, lenvatinib was shown to be as effective as sorafenib; OS of the patients treated with lenvatinib was similar to that of those treated with sorafenib, and lenvatinib was significantly better than sorafenib in terms of antitumor effects such as the response rate, disease control rate, and progression-free survival (PFS) [8]. Second-line therapies for unresectable HCC intolerant to sorafenib or lenvatinib include regorafenib, cabozantinib, sunitinib, linifanib, brivanib, and tivantinib, which target tyrosine kinases and related pathways [125]. However, these molecular-targeted therapies show the development of severe adverse effects, drug resistance, and cytostatic properties, which hinder therapeutic benefits and patient acceptability [126].

As the variety of treatment options increases, the evidence for markers that predict the therapeutic efficacy of MTAs and prognosis for HCC patients treated with MTAs, including sorafenib, becomes more important. This information can help clinicians choose from new therapeutic options available [127]. There is some evidence of biological prognostic markers for sorafenib treatment that has been most clinically used in advanced HCC. Angiogenetic markers, including angiopoietin-2 (Ang-2) and VEGF, have been analyzed in several studies. In the SHARP study, baseline VEGF and Ang-2 plasma levels could not be used as prognostic biomarkers for HCC patients treated with sorafenib and a placebo [128]. In contrast, it was reported that decreases in plasma VEGF levels due to sorafenib treatment after 8 weeks was a predictor of better median OS [129]. Another study showed that HCC patients who were refractory to sorafenib treatment had higher circulating cell-free DNA (cfDNA) levels of VEGF and that the group with cfDNA levels of VEGF above the median had worse time to progression (TTP) [130]. A study evaluating SNPs of multiple angiogenic molecules, including HIF-1α, VEGF, and Ang-2, demonstrated that rs12434438 of HIF-1α, rs2010963 of VEGF-A, and rs4604006 of VEGF-C were dependent factors predicting the therapeutic effects of sorafenib [131]. In the inflammatory TME, circulating immune cells and cytokines also play important roles in HCC progression and could be prognostic markers for sorafenib treatment [127]. A high neutrophil–lymphocyte ratio (NRL) has been reported to be a useful biomarker for predicting poor therapeutic effect of sorafenib [132], and a high NLR before sorafenib treatment is reported to be associated with short OS [133]. Insulin-like growth factor 1 (IGF-1) also has the potential to predict the therapeutic effect of MTAs, including sorafenib. It has been reported that HCC patients with high baseline IGF-1 achieve higher disease control rates and show longer PFS and OS [134].

Immunotherapy is attracting attention as the fourth line of treatment following surgery, radiation therapy, and chemotherapy for cancer and has been elevated to first-line treatment in various cancers. Immune checkpoints are a specific subtype of membrane-bound molecules that act as important regulators of immune escape in cancer by blocking T cell activation and promoting T cell exhaustion. ICIs, some currently approved or under clinical trials for HCC, are primary targets of PD-1/PD-L1 and CTLA-4 [135]. Similar to other direct immunotherapies, adoptive cell transfer-based therapies (ACTs) target ex vivo genetic modification of autologous immune cells [136]. Monotherapy with ICIs or ACTs for HCC has failed to meet the primary clinical endpoints, including decreased tumor size and antitumor responses, due to heterogeneity of tumors, altered TME, drug resistance development, hypervascularity, hypoxia, and severe side effects [137,138]. Besides those in HCC, the response rate of monotherapy with ICIs or ACTs is low in many other cancer types. It is necessary to develop markers that can select cases expected to benefit from therapeutic effects before or early in treatment and improve the therapeutic effects of combination therapy with other molecular-targeted agents.

Other potential therapeutic targets for immune checkpoint molecules include Tim-3, a transmembrane protein expressed in IFN-γ-secreting Th1 cells, NK cells, and CTLs [139]. Tim-3 negatively regulates T cell responses by interacting with galectin-9 (Gal-9). It has been reported that the Tim-3/Gal-9 pathway is consistently associated with poor prognosis in HBV-related HCC patients [140]. A member of the immunoglobulin superfamily of proteins, lymphocyte activation gene 3 protein (LAG-3) represses the co-stimulatory function of T cells by binding to MHC class II molecules and may also be a therapeutic target for HCC. Recently, the immunosuppressive roles of Tim-3 and LAG-3 have been elucidated, although their clinical values need further exploration [141].

## 6. Immune-Based Therapy for HCC

### 6.1. Immune Checkpoint Inhibitors (ICIs)

Immunotherapy with ICIs has changed the treatment paradigm for various cancer types and improved outcomes for patients with advanced or metastatic cancer. CTLA-4 was first cloned in 1987 as a novel molecule expressed on activated T cells [142] and was later shown to have an inhibitory function [143]; in 1996, it was found that the CTLA-4 antibody could suppress cancer growth in the mouse model [144]. As a clinical application of the CTLA-4 antibody, ipilimumab was shown in 2010 to improve the long-term prognosis of patients with unresectable malignant melanoma [145]. PD-1 was first cloned in 1992 [146] and later shown to be a negative regulator of the immune system [147], and in 2002, an antibody against PD-L1, a ligand of PD-1, showed therapeutic effect on cancers in a mouse model [148]. As a clinical application of the PD-1 antibody, nivolumab was shown to be effective against unresectable malignant melanoma [149]. After long development processes, ICIs have recently been widely used for the treatment of melanoma, non-small cell lung cancer, renal cancer, and hematologic malignancy, not only for metastatic cases but also in the adjuvant setting. HCC has a background of chronic liver inflammation such as viral hepatitis, alcoholic hepatitis, and steatosis; therefore, treatment by regulating inflammation and immune response is also effective [150]. PD-1 and CTLA-4 expression is known to be elevated in viral antigen-specific T cells in patients with chronic HCV infection [151], and ICIs are thought to be promising not only for HCV treatment but also for HCC treatment.

In a phase II trial evaluating the effect of tremelimumab (CTLA-4 antibody) in HCV-related HCC patients, the antitumor effect was remarkable, with a partial response (PR) rate of 17.6% and TTP of 6.5 months [152]. In a phase II trial that evaluated the effect of adding radiofrequency ablation or transarterial chemoembolization as adjuvant therapy to enhance the immunogenicity effect, the PR rate was 26.3%, and the TTP was 7.4 months, confirming the effectiveness of tremelimumab against HCC [153].

Nivolumab is the world’s first recombinant human IgG4 monoclonal antibody for PD-1. In the phase I/II dose escalation and expansion trial (CheckMate 040) of nivolumab, 214 HCC patients showed a response rate of 19.6% and OS of 15.5% [154]. Interestingly, the effects of nivolumab persisted in responders. The updated results reported in 2018 at the American Society of Clinical Oncology GI annual meeting (ASCO-GI) indicated that nivolumab had a significant effect in HCC patients, with an OS of 28.6 months and 15 months for first-line and second-line therapies, respectively. In a phase III trial (CheckMate 459) comparing nivolumab with sorafenib, the standard MTA for HCC, the results failed to meet the predefined statistical significance threshold, whereas the OS rate with nivolumab was higher than that with sorafenib [155]. Another study comparing nivolumab as a second-line treatment after sorafenib failure showed no significant difference in survival outcomes [156]. However, nivolumab may be more effective than regorafenib in non-progressors. It was reported that nivolumab treatment improved OS and overall response rate (ORR) compared to that with regorafenib in patients with HCC after sorafenib failure [157]. There are several other ongoing trials on nivolumab (Table 2).

Pembrolizumab is a strong and selective monoclonal antibody of the IgG 4/κ isotype. In a phase II trial (KEYNOTE-244) of pembrolizumab (anti-PD-1), 104 HCC patients showed outcomes similar to those with nivolumab, with a response rate of 17.3% and OS of 12.9 months [159]. However, in phase III trials (KEYNOTE-240), pembrolizumab treatment did not reach specific endpoints of improvement in ORR and PFS when compared with that from sorafenib treatment [160]. The results also indicated that the ORR of pembrolizumab treatment was remarkably higher than that of the placebo group. There are several other ongoing trials on pembrolizumab (Table 2).

Recent studies indicate that ICIs have no effect on patients with WNT/β-catenin-mutated HCC, an immune exclusion subclass [161,162,163]. It has also been reported that the effectiveness of MTAs such as sorafenib is not related to WNT/β-catenin mutation status [163]. Although the frequency of WNT/β-catenin-mutated HCC, 20–30%, is not high [161,162], MTAs rather than ICIs may be suitable as a first-line treatment for WNT/β-catenin-mutated HCC. Furthermore, WNT/β-catenin-mutated HCC shows an iso-high intensity in the hepatobiliary phase of gadolinium ethoxybenzyl diethylenetriaminepentaacetic acid-enhanced magnetic resonance imaging (Gd-EOB-DTPA-MRI: EOB-MRI) [164]. EOB-MRI may be useful for predicting the state of WNT/β-catenin mutation in HCC and treatment resistance to ICIs.

### 6.2. Combination Therapy Including ICIs

The usefulness of angiogenesis inhibitors has been confirmed in HCC treatment, and it has been pointed out that synergistic effects due to the immunosuppressive action of VEGF/VEGFR can be expected. The VEGF/VEGFR-targeted therapies are often selected in combination with ICIs [165]. Direct effects on immune cells include inhibition of differentiation and maturation of effector T cells and DCs, enhanced expression of immune checkpoint molecules, and proliferation and tumor migration of Tregs and MDSCs. Indirect effects include decreased expression of adhesion molecules such as intracellular adhesion molecule-1 (ICAM-1) and vascular cell adhesion molecule-1 (VCAM-1) on vascular endothelial cells, the presence of a hypoxic environment due to fragile angiogenesis, and decreased immune cell infiltration.

Atezolizumab is a monoclonal antibody against PD-L1. PD-L1 inhibits the immune system by inducing the proliferation of antigen-specific CD8+ cells and regulating the accumulation of exogenous antigen-specific T cells [166]. The phase III open-label study (IMbrave 150) in advanced HCC patients treated with combination atezo+bev, a humanized anti-VEGF monoclonal antibody, demonstrated that the hazard ratio for death with atezo+bev as compared with sorafenib was 0.58 in the primary analysis. Additional results showed that the OS rate at 12 months was 67.2% vs. 54.6%, PFS was 6.8 months vs. 4.3 months (atezo+bev vs. sorafenib, respectively). The incidence of grade 3 or higher adverse events was 56.5% in the atezo+bev group and 55.1% in the sorafenib group [12]. With these better outcomes, atezo+bev combination therapy has rapidly found clinical application as a first-line treatment for advanced HCC and has replaced sorafenib.

Combination therapies with MTAs and ICIs may play central roles in HCC treatment, and clinical trials have reported the effect of adding PD-L1 antibodies to lenvatinib [15,167,168]. In a phase 1b trial with pembrolizumab plus lenvatinib treatment for HCC patients with Child–Pugh A, favorable results were achieved, with a median PS of 22.0 months, PFS of 9.3 months, and ORR per modified response evaluation criteria in solid tumors (mRECIST) of 46.0% [169]. A phase III trial (LEAP 002) comparing lenvatinib plus pembrolizumab combination therapy with lenvatinib monotherapy is currently ongoing, and additional evidence of the efficacy of combination therapy is expected (Table 3). In 2020 at ASCO, combination therapy with durvalumab (PD-L1 antibody) and tremelimumab showed promising outcomes, with a median OS of 18.7 months, median PFS of 2.2 months, and ORR per RECIST v1.1 of 24.0% [14]. Combination therapy with durvalumab and tremelimumab is likely promising, and a phase III trial (HIMALAYA) is currently ongoing (Table 3). Combination therapy with nivolumab and ipilimumab (CTLA-4 antibody) is reportedly effective in patients with advanced HCC. In a phase I/II trial (CheckMate 040), this combination therapy showed a median OS of 22.8 months, and ORR per RECIST v1.1 was 32% [167]. A phase III trial (CheckMate 9DW) comparing the combination of nivolumab and ipilimumab with lenvatinib or sorafenib is currently ongoing (Table 3).

The positive results of the IMBrave trial have positioned combination atezo+bev as the first-line treatment for advanced HCC, but subsequent sequential MTA therapy is still important. It has been reported that the binding of nivolumab to CD8+ T cells lasts for more than 20 months in treatment for lung cancer [170]. Even in cases where the PD-1/PD-L1 antibody leads to progressive disease, post-treatment with MTAs changes the TME from immune suppressive to immune permissive. Sequential ICI–MTA therapy is expected to bring about TME changes similar to those from PD-1/PD-L1 antibody plus MTA combination therapy and to show a favorable therapeutic effect [171,172].

### 6.3. Adoptive Cell Transfer-Based Therapies (ACTs) in HCC

ACT offers durable antitumor immunity, and the exploration and development of immunotherapies for the treatment of HCC have attracted considerable attention. Recent research and clinical case reports have indicated the success of engineered autologous HBV-specific TCR redirected therapeutics for HBV-related HCC [173,174]. ACT-based therapies for HCC include cytokine-induced killer (CIK) cell treatment and chimeric antigen receptor-modified (CAR) T cells. The main ACTs are presented in Table 4.

CIK cells are a mixture of cells that proliferate in the presence of cytokines such as IFN-γ, IL-1, and IL-2 and comprise activated NKT cells, CD3-/CD56+ NK cells, and CD3+/CD56- CTLs [175]. It has been suggested that CIK cells suppress HCC progression by effectively killing cancer stem cells (CSCs) through natural killer group 2 member D (NKG2D)-ligand recognition. A recent study indicated that a high number of PD-1+ TILs is positively correlated with favorable outcomes of CIK cell treatment for HCC [176]. In a phase III trial examining the effectiveness of CIK cell treatment as adjuvant therapy in patients who underwent curative treatment for HCC, activated CIK cell treatment was created by incubating patients’ peripheral blood mononuclear cells with IL-2 and an antibody against CD3. CKI cell treatment prolonged recurrence-free survival (RFS) and OS; RFS was 44.0 months vs. 30.0 months (immunotherapy group vs. control group) [177]. In addition, the proportion of patients with serious adverse events did not differ significantly between the groups. Based on the evidence obtained from the growing literature on HCC, CIK cell treatment can be a promising adoptive immunotherapy to prevent recurrence or as an adjuvant treatment for HCC.

CAR-T therapy is a promising strategy for HCC treatment that specifically recognizes TAAs and effectively eliminates tumor cells in a non-MHC manner. Although CAR-T therapy has shown excellent therapeutic effects in hematological malignancies, the results of CAR-T therapy for solid tumors, especially HCC, are still modest. A recent study indicated that glypican-3 (GPC-3), an oncofetal proteoglycan anchored to the HCC cell membrane and associated with HCC progression, would provide a novel immunotherapeutic target in the treatment of HCC [178]. In previous experiments involving xenograft models of human HCC, GPC3-CAR-T cells efficiently suppressed tumor growth and impressively eradicated tumor cells that highly expressed GPC3 proteins [179]. In another in vitro study, exposure of HCC cells to GPC3-specific NK cells resulted in significant cytotoxicity and cytokine production. The potent antitumor activity of GPC3-specific NK cells was observed in multiple HCC xenografts with both high and low GPC3 expression but not in those without GPC3 expression [180], which extends the treatment options for HCC patients with high GPC-3 expression. In summary, CAR-T therapy provides new avenues for HCC immunotherapies, and further elucidation of the clinical efficacy and on-target off-tumor toxicity is required before its widespread clinical application.

### 6.4. Non-Cell-Based Vaccine and Oncolytic Viruse-Based Immunotherapy in HCC

Vaccines targeting TAAs have been developed in HCC vaccine-based therapy to identify a growing number of TAAs, including alpha-fetoprotein (AFP), GPC-3, and telomerase-reverse transcriptase (TERT). The first clinical trial for an HCC AFP-vaccine was completed with only a transient immunological response detected due to deficient CD4+ helper T cell support [181]. Partial clinical use of GPC-3-based vaccines resulted in the induction of a measurable antitumor response associated with prolonged OS in HCC patients [182]. In contrast, the hTERT-derived peptide vaccine, which binds multiple HLA class II molecules, resulted in little clinical activity and no absolute antigen-specific CTL response [183]. Besides these classical TAAs, cancer-testis antigens (CTAs) are attractive targets for HCC immunotherapy [184]. Expression of NY-ESCO-1, one of the most immunogenic CTAs, is associated with poor outcomes, and DCs loaded with the NY-ESO-1 peptide could stimulate specific T cell responses to HCC cells in vitro [185]. The main non-cell vaccines are summarized in Table 4.

Oncolytic viruses (OVs) selectively replicate in tumor cells, resulting in their damage without harm to normal cells, and may contribute to the development of new strategies for HCC immunotherapy. Recent reports have shown that ICIs are associated with nonspecific T cell activation and may be effective in combination therapy with OVs [186]. As OVs are used for HCC immunotherapy, the therapeutic gene recombinant oncolytic adeno-associated viruses (AAVs) can exert a strong cytopathic effect on HCC. As another recombinant AAV model, AAV vectors containing the hTERT and TRAIL genes target the telomerase activity of HCC cells and exhibit specific cytotoxicity and apoptosis to suppress HCC growth [187].

## 7. Immune-Related Adverse Events (irAE) in HCC

While ICIs inhibit negative feedback mechanisms of the immune system resulting very durable antitumor responses through blockade of CTLA-4 or PD-1, these immune-related adverse events (irAEs) can interfere with the immunosuppressive therapy. ICIs can influence peripheral tolerance to autoantigens, resulting in antibody formation, which causes damage to various organs. ICIs also release T cells, with subsequent production of pro-inflammatory cytokines such as IFN-γ and TNF, resulting in excessive autoimmunity tumor inflammation. The onset of irAEs is unpredictable because patients can experience irAEs early after ICI treatment up to more than 18 months after initiation of treatment [188] and may develop multiple different irAEs either simultaneously or subsequently [189]. The frequency of irAEs has been reported to be 60–85% in patients treated with CTLA-4 antibodies, 57–85% in patients treated with anti-PD-1, and 95% in patients treated with the combination of anti-CTLA-4 and anti-PD-1 [190]. While most irAEs are reversible and can be addressed, it is important to recognize fatal irAEs. According to meta-analysis of anti-CTLA-4 or anti-PD-L1 monotherapy, common target organs with irAEs were the skin, gastrointestinal tract, endocrine glands, and the liver [189,191]. The most frequent fatal irAEs were pneumonitis, hepatitis, and neurotoxic effects [189]. Combination therapy with anti-CTLA-4 and anti-PD-(L)1 is prone to appear incrementally toxic. Patients treated with anti-CTLA-4 and anti-PD-(L)1 combination therapy had a higher risk and more frequent multi-organ involvement than ICI monotherapy [192]. The incidence rates of grade 3 or higher in the Common Terminology of Clinical Adverse Events (CTCAE) categorization was 41%, and the common fatal events were myocarditis, myositis, and neurologic events [192]. Although fatal irAEs remains rare, irAEs should be recognized promptly as early interventions may alleviate future complication.

In general, the incidence of irAEs in HCC patients treated with ICIs such as nivolumab, pembrolizumab, and ipilimumab is not significantly different from other cancer types. The incidence of severe irAEs tended to be lower in HCC, ranging from 10–20%, compared to more than 30% in other cancers [193]. The incidence of hepatic irAEs is higher in HCC patients, possibly due to the unique liver immunobiology and chronic inflammatory conditions of the liver such as cirrhosis and viral hepatitis. While only 3% of patients experienced hepatitis enzyme elevation and 0.85% of patients experienced treatment-related hepatitis among other cancer types [194], 13.7% of HCC patients treated with anti-PD-1 monotherapy experienced hepatitis enzyme elevation of any grade [195]. Cutaneous toxicity is the first common irAE among patients treated with ICIs, the incidence of rash and pruritus in HCC patients treated with nivolumab and pembrolizumab monotherapy was 11–23% and 10-11.5%, and 13–19% and 12–18.3%, respectively [196]. As for digestive system irAEs, diarrhea is also a frequent adverse reaction due to ICIs. Diarrhea occurred in 10% of HCC patients treated with nivolumab, 11% treated with pembrolizumab, and 12–24% treated with nivolumab and ipilimumab combination therapy [196]. Colitis is a serious irAE that can cause fatal colonic perforation and peritonitis. Most ICI-induced colitis in HCC patients is grade 3 and occurs in 1% of those treated with PD-L1 antibodies and 2.6% of those treated with a CTLA-4 antibody and PD-L1 antibody combination [196]. The incidence of irAEs among HCC patients treated with ICIs was relatively low in the other targeted organs. The incidence of pneumonia in HCC patients treated with nivolumab was approximately 3%, and there were few patients with events grade 3 or higher. In a phase II pembrolizumab clinical trial, 7% of HCC patients reported myalgia, but no patients were found in the phase III trial [196]. In other ICI trials for HCC, the incidence of heart-related complications was extremely rare, less than 1%. In clinical trials of tremelimumab for HCC, 15% of those patients developed grade 3 or higher encephalopathy, yet these complications are more associated with underlying cirrhosis than ICI treatment. The incidence of ICI-induced myositis was also rare, and the frequency during ICI treatment for HCC is unknown, but it is a cautionary irAE that can develop suddenly and progress rapidly [197]. Despite the low incidence, if left untreated, complications related to the respiratory, circulatory, and nervous systems can be fatal. For rare irAEs, early diagnosis and proper management are important to improving the outcome of ICI treatment.

As VEGF-targeted therapies can facilitate drug delivery to HCC tissues, combination with ICIs and VEGF inhibitors may reduce the dose of ICIs, thus reducing the toxicity from ICIs compared to ICI monotherapy [198]. According to a phase III clinical trial of atezo+bev combination therapy for HCC, the incidence of all grades and high-grade irAEs in patients treated with an atezo+bev combination was not significantly different from anti-PD-1 monotherapy or a PD-1 antibody plus CTLA-4 antibody combination [12]. According to the assessment of the common irAEs in ICI monotherapy, patients treated with atezo+bev combination therapy had lower incidence of cutaneous irAEs, with 12% rash and 19.5% pruritus. The incidence diarrhea was not significantly different, and less than 10% of patients reported abnormal thyroid function. The most common adverse event in atezo+bev combination therapy was hypertension with an incidence of 29.8%. In addition, 56.5% of patients treated with this combination had some grade 3 or 4 side effects compared with 55.1% of patients who were treated with sorafenib [12]. The side effects of grade 3 or higher were hypertensive encephalopathy (15.2%), bleeding (7.9%), myelosuppression (8.5%), infection (8.2%), and nephrotic syndrome (3.0%), and most were influenced by bevacizumab. As the use of ICIs continues to expand, early detection and management of irAEs are crucial to maximize the duration of treatment while minimizing toxicities for patients. The safety of ICIs in the setting of advanced liver cirrhosis and their efficacy in different etiologies remain open to questions and needs to be addressed in future trials.

## 8. The Immunological Classification and Biomarkers for HCC Immunotherapy

Although various immunotherapeutic clinical trials have been conducted for HCC, there is insufficient evidence of the efficacy of HCC biomarkers as indicators of therapeutic effects and prognosis. In general, PD-1/PD-L1 expression, microsatellite instability, and tumor mutation burden are used as biomarkers associated with immunotherapeutic effects in various cancer types. In HCC, clinical trials with nivolumab (CheckMate 040) showed that the response rate was 18.6% with PD-L1 expression of less than 1%, and there was no significant difference depending on PD-L1 expression [154]. Microsatellite instability is rare in HCC, which may interfere with its use as a biomarker [199]. It has been reported that the mutation of AT-rich interaction domain 2 (ARID2), which is a constituent of the SW1/SNF complex, might be a tumor mutation burden in HCC and may correlate with the efficacy of ICIs; however, it has not yet reached the stage of clinical application [200].

A recent gene expression analysis of HCC showed that there are immune classes, cases in which the inflammatory response is strong and the PD-1/PD-L1 pathway is activated. The immune class is further classified into a group with many CD8+ T cells (active) and a group with many M2 macrophages by activating the TGF-β pathway (exhausted), and a good prognosis has been found only in the active immune class [201]. In another study using multiple immunostaining procedures, HCC was classified into three groups (immune-high, -mid, and -low) according to the degree of inflammatory cell infiltration, and it was found that the immune-high group had a good prognosis with infiltration of many B cells, plasma cells, and T cells [202]. It was also clarified that there were many cases of strong HCC fatty degeneration in the immune-mid group, the Wnt/β-catenin pathway was activated in the immune-low group, and the prognosis was poor in cases with many Tregs. Further up-to-date HCC gene expression and exosome analysis suggested that HCC can be categorized into the MS1 group with TP53 mutation, a robust cancer cell proliferation, and chromosomal instability, which were associated with poor prognosis; the MS2 group with catenin β1 (CTNNB1) mutation and high DNA methylation level; and the MS3 group, which was strongly associated with metabolic syndrome [203]. It was reported that cancer immunity was strongly suppressed in the MS2 group. Furthermore, in the MS3 group, there was a good prognosis group (MS3i), which had a strong inflammatory response and activated PD-1/PD-L1 pathway. Taken together, these studies show that activation of the PD-1/PD-L1 pathway during HCC has a good prognosis and that the CTNNB1 mutation suppresses cancer immunity. It has also been reported that ICIs are ineffective against HCC with CTNNB1 and AX1N1 mutations [163]. Thus, it is necessary to make better utilization of preclinical studies using immunological classification and animal models to solve the remaining issues such as predicting efficacy factors or optimizing combinations of anticancer drugs and MTAs.

## 9. Conclusions

The unique characteristics of hepatic immunity contribute to oncogenesis and tumor tolerance in chronic inflammation and fibrosis in the liver. Immunosuppressive cells such as MDSCs, TAMs, TANs, CAFs, and Tregs are the key components of TMEs that promote HCC growth and invasion. The differentiation and maturation of these immune cells require the regulation of several cytokines involved in TMEs in addition to the interaction of receptors and related ligands. In recent years, various treatment options for HCC, especially MTAs, have been applied clinically; however, HCC still has one of the worst prognoses, and a new treatment strategy targeting immunity is required. Due to the immunological properties of HCC, there is an opportunity to adapt immunotherapy, including ICIs, in HCC. In advanced HCC and other cancer types, immunotherapeutic approaches have increasingly focused on monoclonal antibodies against CTLA-4 and PD-1 that block ICI pathways. As this review showed, positive results of HCC immunosuppressive treatment, including atezo+bev combination therapy, has resulted in a paradigm change in the treatment strategy for advanced HCC. It is believed the publication of results of ongoing clinical trials with immunosuppressive therapies for advanced HCC, including various ICIs, will provide further insights into effective treatment. To further maximize immunosuppressive treatment and improve the prognosis of HCC patients, the accumulation of detailed information involving paired tumor biopsy will become more important in the clinical course of HCC immunotherapy. In addition, advances in DNA and RNA sequencing technologies will provide evidence for the mechanisms underlying HCC development to identify therapeutic targets.

## Figures and Tables

**Figure 1 ijms-22-05801-f001:**
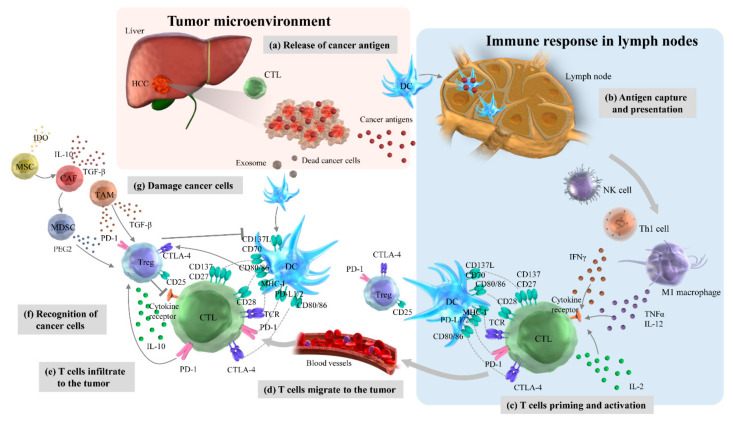
Seven steps of the cancer-immunity cycle in hepatocellular carcinoma (HCC). (**a**) Cancer antigens are released from HCC cells that have died due to proliferation necrosis or treatment. (**b**) Dendritic cells (DCs) that capture cancer antigens or phagocytose dead HCC cells migrate to the lymph node. (**c**) DCs digest cancer antigens, place them on MHC class I molecules as antigen peptides, and present them to CD8+ cytotoxic T lymphocytes (CTLs) in the lymph node. (**d**) T cells migrate to HCC tissue. (**e**) T cells infiltrate to the tumor. (**f**) T cells recognize HCC cells via T cell receptors (TCR). The co-stimulatory receptors CD28 and CD137 bind to ligands CD80/CD86 and CD137L on DCs, respectively. (**g**) T cells kill HCC cells. Inflammatory cytokines tumor necrosis factor (TNF)-β and interleukin (IL)-12 induced by M1 macrophages, interferon (IFN)-γ produced by Th1 and natural killer (NK) cells, and IL-2 secreted by CTLs fuel further T cell activation. Checkpoint molecules CTL-associated protein-4 (CTLA-4) and programmed cell death protein-1 (PD-1) bind to CD80/CD86 and PD-ligand 1 (PD-L1)/PD-L2 on DCs, respectively, suppress T cells, and control excessive immune response (right part of Figure 1). The immunity cycle is equipped with a negative feedback mechanism; when excessively suppressed, the cycle is stopped along with the amplification of the cancer immune response. Even if immunosuppressive factors and cells become significant and tumor microenvironment (TME) is established, this cycle is stopped, and the immune response is reduced. Epithelial-mesenchymal transition (EMT) in HCC cells induces differentiation of myeloid-derived suppressor cells (MDSCs), tumor-associated macrophages (TAMs), and regulatory T cells (Tregs) by mesenchyme stem cells (MSCs) and cancer-associated fibroblasts (CAFs). Immunosuppressive cytokines such as IL-10, transforming growth factor (TGF)-β, indoleamine 2,3-dioxygenase (IDO), and prostaglandin E2 (PGE2) are also involved, and the expression of PD-1 on CTLs and PD-L1/PD-L2 on DCs is increased. T cell activation is further attenuated through direct suppression of Tregs against CTLs and trans-endocytosis of CD80/CD86 on DCs by Tregs (left side of Figure 1).

**Table 1 ijms-22-05801-t001:** Summary of immune cells and molecules in the tumor immune environment of HCC.

Immune Cell	Molecule(s)	Major Effects	Reference(s)
MDSC	GM-CSF, IL-β, IL-6, VEGF, MCP-1	MDSC accumulation and migration adjustment	[65]
	EZH2 enzyme	Activation of NF-κB Accumulation of polymorphonuclear MDSCs	[66]
	CCL26	Invasion to the hypoxic region of HCC tissues	[67]
	ENTPD2	MDSC accumulation by converting extracellular ATP to 5′ AMP	[68]
TAM	CCL17, CCL18, CCL22	Attraction of Treg cells to HCC tissues Activation of CTLs	[69,70]
	IL-1β	Promoting EMT and HCC immune evasion	[71]
	TNF-α, IL-β, IL-6, IL-23	Expansion of IL-17-producing CD4+ Th17 cells	[72]
	TGF-β	Promotion of TIM-3 expression in TAMs	[73]
TAN	CXCL5	TAN infiltration, indicating a poor prognosis	[74]
	CCL2, CCL17	Associated with tumor size, microvascular invasion, differentiation, and clinical stage	[75]
	HIF-1α	TAN infiltration associated with HCC progression	[75]
CAF	IDO, PGE2	TNF-α and INF-γ production by NK cells Associated with HCC development	[76]
	BNP-4	Activation of hepatic fibroblasts Enhancing CAF invasiveness	[77]
Treg	AP-1, NFAT1	Promotion of immunosuppression in HCC	[78]
	TGF-β	Treg infiltration into the liver	[79]
TIL	FOXP3	Master transcription factor of Treg Associated with poor prognosis	[80,81]
CTL	VEGF, CXCL17, IL-10, IDO	Causing poor production of INF-γ by CTLs	[67,82,83,84]
	IL-2, IDO	CTL suppression	[85]

MDCS, myeloid-derived suppressor cell; TAM, tumor-associated fibrosis; TAN, tumor-associated neutrophil; CAF, cancer-associated fibroblast; Treg, regulatory T cell; TIL, tumor-infiltrating lymphocyte; CTL, CD8+ cytotoxic T lymphocyte; GM-CSF, granulocyte-macrophage colony-stimulating factor; IL-β, interleukin-β; VEGF, vascular endothelial growth factor; MCP-1, monocyte chemotactic protein-1; EZH2, enhancing the zeste homolog 2; CCL26, C-C motif ligand 26; ENTPD2, ectonucleoside triphosphate diphosphohydrolase 2; TNF-α, tumor necrosis factor α; TGF-β, transforming growth factor β; CXCL5, C-X-C motif ligand 5; HIF-1α, hypoxia-inducible factor 1α; IDO, indoleamine 2,3-dioxgenase; PGE, prostaglandin E2; BNP-4, bone morphogenetic protein 4; AP-1, activator protein 1; NFAT1, nuclear factors of activated T cell 1cytotoxic T lymphocyte-associated protein 4; FOXP3, forkhead box P3; NF-κB, nuclear factor-κB; EMT, epithelial-mesenchymal transition.

**Table 2 ijms-22-05801-t002:** Summary of immune checkpoint inhibitors (ICIs) for advanced hepatocellular carcinoma (HCC).

Regimen	Target Population	Design	Registration Number	Result	Reference
Anti-CTLA-4 antibody					
Tremelimumab	Second-line for advanced HCC	Single group assignment	NCT01008358	Effective	[152]
Tremelimumab	Neoadjuvant before ablation	Single group assignment	NCT01853618	Effective	[153]
Anti-PD-1 antibody					
Nivolumab	First- and second-line for advanced HCC	Single group assignment	NCT01658878	Positive	[154]
Nivolumab	First-line for advanced HCC	Versus sorafenib	NCT02576509	Negative	[155]
Nivolumab	Adjuvant after curative treatment	Versus placebo	NCT03383458	Ongoing	
Nivolumab	Improvement after TACE	Versus placebo	NCT04268888	Ongoing	
Tislelizumab	First-line for unresectable HCC	Versus sorafenib	NCT03412773	Ongoing	[158]
Anti-PD-L1 antibody					
Pembrolizumab	Second-line for advanced HCC	Single group assignment	NCT02702414	Effective	[159]
Pembrolizumab	Second-line for advanced HCC	Versus placebo	NCT02702401	Negative	[160]
Pembrolizumab	Adjuvant after curative treatment	Versus placebo	NCT03867084	Ongoing	

CTLA-4, cytotoxic T lymphocyte-associated protein 4; PD-1, programmed cell death protein 1; PD-L1, PD-ligand 1.

**Table 3 ijms-22-05801-t003:** Summary of combination therapy, including immune checkpoint inhibitors (ICIs).

Regimen	Target Population	Design	Registration Number	Result	Reference
Anti-CTLA-4 antibody-based Combination therapy					
Tremelimumab + TACE	First-line for advanced HCC	Single group assignment	NCT01853618	Effective	[168]
Anti-PD-1 antibody-based Combination therapy					
Nivolumab + cabozantinib	Neoadjuvant before resection	Single group assignment	NCT03299946	Ongoing	
Nivolumab + ipilimumab	First-line for advanced HCC	Versus lenvatinib or sorafenib	NCT04039607	Ongoing	
Nivolumab following SIRT	Intermediate HCC	Single group assignment	NCT03380130	Ongoing	
Nivolumab following resection	First diagnosis of HCC	Versus placebo	NCT03383458	Ongoing	
Nivolumab + galunisertib	Second-line for advanced HCC	Single group assignment	NCT02423343	Ongoing	
Nivolumab + lenvatinib	First-line for advanced HCC	Single group assignment	NCT03418922	Ongoing	
Nivolumab + sorafenib	Advanced HCC	Single group assignment	NCT03439891	Ongoing	
Nivolumab + DEB-TACE	Intermediate HCC	Parallel assignment	NCT03143270	Ongoing	
Nivolumab + mogamulizumab	Second-line for advanced HCC	Single group assignment	NCT02705105	Ongoing	
Nivolumab + relatlimab	Advanced HCC	Parallel assignment	NCT01968109	Ongoing	
Anti-PD-L1 antibody-based Combination therapy					
Atezolizumab + bevacizumab	First-line for advanced HCC	Versus sorafenib	NCT03434379	Positive	[12]
Atezolizumab + cabozantinib	First-line for advanced HCC	Versus sorafenib	MCT03755791	Ongoing	
Durvalumab + tremelimumab	First-line for advanced HCC	Versus sorafenib	NCT03298451	Ongoing	
Durvalumab + ramucirumab	Second-line for advanced HCC	Parallel assignment	NCT02572687	Ongoing	

TACE, transcatheter arterial chemoembolization; SIRT, selective internal radiation therapy; DEB-TACE, drug-eluting bead transarterial chemoembolization.

**Table 4 ijms-22-05801-t004:** Summary of adoptive cell transfer-based therapies (ACTs) for advanced hepatocellular carcinoma (HCC).

Regimen	Target Population	Design	Registration Number
CIK monotherapy			
CIKs	HCC	Phase III clinical trial	NCT00769106
CIKs	HCC, RCC, and lung cancer	Phase I clinical trial	NCT01914263
DC-CIKs	HCC	Phase III clinical trial	NCT01821482
CIK-based Combination therapy			
CIKs + PD-1 antibodies	HCC, RCC, bladder cancer, Colorectal cancer, and NSCLC	Phase II clinical trial	NCT02886897
CIKs + TACE	HCC	Phase III clinical trial	NCT02487017
CIKs + RFA	HCC	Phase III clinical trial	NCT02678013
CAR-T trials			
Anti-GPC3 CAR-T	HCC	Phase I/II clinical trial	NCT03084380
Anti-GPC3 CAR-T	HCC	Phase I/II clinical trial	NCT02723942
Anti-GPC3 CAR-T	AFP-expressing HCC	Phase I clinical trial	NCT03349255
Anti-GPC3 CAR-T	Advanced HCC	Phase I clinical trial	NCT03198546
TAI-GP3-CAR-T	HCC	Phase I/II clinical trial	NCT02715362
Anti-Mucin1 CAR-T	HCC, NSCLC, pancreatic cancer, and triple-negative breast cancer	Phase I/II clinical trial	NCT02587689
CAR-T targeting TAAs	HCC, pancreatic cancer, and colorectal cancer	Phase I/II clinical trial	NCT02959151

CIK, cytokine-induced killer; RCC, renal cell carcinoma; DC-KICs, dendric and cytokine-induced killer cells; TACR, transcatheter arterial chemoembolization; RFA, radiofrequency ablation; CAR-T, cells chimeric antigen receptor-T cells; GPC3, glypican 3; TAI, transcatheter arterial infusion.

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
