# Peer review of "Tumor Immune Microenvironment and Immunosuppressive Therapy in Hepatocellular Carcinoma: A Review"

_ijms, 2021, doi:10.3390/ijms22115801_

Round 1

Reviewer 1 Report

Kyoko Oura uncovered HCC the tumor microenvironment (TMA) available data to be relevant in order to sketch and review the modern concept of immunotherapy for HCC.

The manuscript is of interest, nonetheless, few points need to be addressed.

1) The rationale of why the authors came up with this review.

2) What is the information that is not exactly available that motivated the authors to come up with this information. What are the current caveats and how do the authors highlight the current research in answering them? If not they need to address it in future directions.

3)the manuscript should be amended to provide a new sub-section about the side effects of immune checkpoint inhibitors. Moreover, I think that a short resume of history research on immune checkpoint inhibitors will be useful for readers.

4) This reviewer personally misses some important findings regarding predictive and prognostic factors in HCC patients treated with sorafenib. Indeed, unlike other targeted therapies, predictive and prognostic markers in HCC patients treated with sorafenib are lacking. Their identification could help clinicians in the daily management of these patients, mostly in light of the new therapeutic options available in the first (please refer to PMID: 31640191).

5) The authors need to come up with a table to show the role of TME cells as Checkpoint For Immunological  Patrolling in HCC to highlight their exact role.

6) The authors need to highlight what new information the review is providing to enhance the research in progress.

Author Response

  1. The rationale of why the authors came up with this review.

Response: Thank you very much for your comments.

Although the systemic therapies for advanced hepatocellular carcinoma (HCC) has undergone a major paradigm shift, the results of treatment for advanced HCC remains inadequate due to the lack of evidence associated with treatment resistance and prediction of treatment response. Since the efficacy of immunosuppressive therapies including immune check point inhibitors (ICIs) depends on the tumor immune microenvironment, it is necessary to elucidate the immune environment of HCC to select appropriate drugs. We believe that elucidating the TME, especially the immune environment, in HCC development and progression will contribute to the current core therapeutic strategies including ICIs and improve the prognosis of HCC patients. Therefore, in this review, we have summarized and discussed the tumor immune microenvironment and immunosuppressive therapy in HCC.

 Since sorafenib was first shown to prolong the survival of patients with unresectable HCC (Chen AL, et al. Lancet Oncol 10: 25-34, 2009), systemic therapy with molecular targeted agents (MTAs) has continued to evolve significantly as a useful therapeutic strategy for advanced HCC. Several MTAs such as sorafenib, lenvatinib, regorafenib, ramucirumab, and cabozantinib have found widespread clinical applications and have improved the outcome of HCC treatment, but the results are still unsatisfactory. Elucidation of the tumor microenvironment (TME), which is associated with malignant transformation and drug resistance, will further improve the therapeutic outcome of MTAs.

 In addition to MTAs, immune-based therapies including ICIs have become an indispensable treatment strategy in the clinical practice. The combination of ICI and vascular endothelial growth factor (VEGF) inhibitor has shown better results than sorafenib in advanced HCC (Finn R, et al. N Engl J Med 382(20): 1894-1905, 2020), and atezolizumab plus bevacizumab (atezo+bev) combination is now positioned as the first-line therapy for patients with unresectable HCC.

 Suppression of anti-tumor immune responses in TME is intricately related various mechanisms, including not only immune checkpoint molecules but also suppressive cells such as regulatory T cells (Tregs), myeloid derived suppressor cells (MDSCs), and M2 macrophages, and soluble factors such as cytokines and chemokines. In immunosuppressive therapy for HCC, direct effects on immune cells are related to inhabitation of differentiation and maturation of effector T cells and dendritic cells, increased expression of immune checkpoint molecules, and inhibition of proliferation and migration of Tregs and MDSCs. The indirect effects of immunotherapy on HCC contribute to the suppression of tumor growth through various pathways, including decreased expression of intercellular adhesion molecule 1 (ICAM-1) and vascular cell adhesion molecule 1 (VCAM-1) on vascular endothelial cells, the presence of a hypoxic environment due to juvenile vessel growth, and decreased immune cell infiltration. There are still many unanswered questions about the TME and the systemic immune response to immunosuppressive therapies including ICIs, and many studies are underway to investigated the dynamics of TME to discover novel biomarker and therapeutic targets.

 According to your comment, we have altered the corresponding part in the abstract and introduction section to emphasize the rationale for thinking that this review will contribute to HCC treatment strategies.

  1. What is the information that is not exactly available that motivated the authors to come up with this information. What are the current caveats and how do the authors highlight the current research in answering them? If not they need to address it in future directions.

Response: Immunosuppressive therapy is attracting attention as a new therapeutic strategy following surgery, radiation, and chemotherapy. In particular, ICIs such as antibodies against cytotoxic T lymphocyte associated protein 4 (CTLA-4) and programmed cell death protein 1 (PD-1) / PD-L1 have been expanding their indications to various cancer types in recent years, and have become an indispensable treatment strategy in the clinical practice. The combination of ICI and VEGF inhibitor has shown better results than sorafenib in advanced HCC, and atezo+bev combination is now positioned as the first-line therapy for patients with unresectable HCC. Since few articles comprehensively summarize recent findings on TME for immunotherapeutic targets in HCC, we summarize and discusses the tumor immune microenvironment and immunosuppressive therapy in HCC. We believe that elucidation of the TME, especially the immune environment, in the development and progression of HCC will contribute to improving the current core therapeutic strategies including ICIs, and prognosis of advanced HCC patients.

 Currently, multiple MTAs and ICIs have been introduced into routine practice for advanced HCC, and sequential administration of these agents is expected to prolong overall survival, but the appropriate treatment strategy is unclear. Specifically, how to predict ineffective cases and effective cases, and how to optimize including conventional anticancer agents and MTAs are important issues. At present, clinical trials tend to precede the confirmation of immunotherapeutic effects, and it is necessary to utilize pre-clinical studies using cancer immunological classification and animal models in the future.

 According to your comment, we have altered the corresponding part in the introduction to emphasize the rationale for thinking that this review will contribute to immunosuppressive treatment strategies for HCC. We also have added the sentences associated with the immunological classification and biomarkers for HCC immunotherapy in section 7.

  1. The manuscript should be amended to provide a new sub-section about the side effects of immune checkpoint inhibitors. Moreover, I think that a short resume of history research on immune checkpoint inhibitors will be useful for readers.

Response: Despite the clinical efficacy, ICIs induce several immune-related adverse events (irAEs), limiting their use in cancer patients and causing treatment failure. ICIs can influence peripheral tolerance to autoantigens, resulting in antibody formation, which causes damage to various organs. ICIs also release T cells, with subsequent production of pro-inflammatory cytokines such INF-γ and TNF, resulting in excessive of autoimmunity tumor inflammation. In the safety profile, endocrine disorder (including type 1 diabetes, thyroid dysfunction, hypopituitarism, and adrenal insufficiency), interstitial lung disease, digestive system disorders, and nervous system disorders (including severe myasthenia, myasthenia, and neuropathy), are seen as irAEs not found in conventional cell-killing antitumor drugs or MTA (Hussaini S, et al. Cancer Treat Rev 92: 102134, 2021). Notably, it has been reported that hepatotoxicity is an important irAE, occurring in up to 16% of treatment cases (Peeraphatdit T, et al. Hepatology 72(1): 315-329, 2020). Most irAEs are mild and untreated but sometimes causes serious, life-threatening events that require caution (Borghaei H, et al. N Engl J Med 373(17): 1627-1639, 2015). In a phase III clinical trial of atezo+bev combination therapy for HCC, 56.5% of patients treated with atezo+/bev combination had some Grade 3 or 4 side effects compared with 55.1% of patients were treated with sorafenib, although not all were irAEs (Finn R, et al. N Engl J Med 382: 1894-1905). The side effect of Grade 3 or higher were hypertensive encephalopathy (15.2%), bleeding (7.9%), myelosuppression (8.5%), infection (8.2%), and nephrotic syndrome (3.0%), and most were thought to be caused by bevacizumab. This result supports the findings that, even in HCC patients, serious irAEs due to atezolizumab are rare, and most are mild enough to be treated. As the use of ICIs continues to expand, early detection and management of irAEs are crucial to maximize the duration of treatment while minimizing toxicities for patients. According your comment, we have added the sentences associated with side effects of ICIs in section 6.3.

 Immunotherapy with ICIs has changed the treatment paradigm for various cancer types and improved outcomes of patients with advanced or metastatic cancer. CTLA-4 was first cloned in 1987 as a novel molecule expressed on activated T cell (Brunet J, et al. Nature 328: 267-270, 1987) and was later shown to have inhibitory function (Krummel MF, et al. J Exp Med 182: 459-465, 1995); in 1996, it was found that CTLA-4 antibody could suppress cancer growth in the mouse model (Leach DR, et al. Science 271: 1734-1736, 1996). As a clinical application of CTLA-4 antibody, ipilimab was shown in 2010 to improve the long-term prognosis of patients with unresectable malignant melanoma (Hodi FS, et al. N Engl J Med 363: 711-723, 2010). PD-1 was first cloned in 1992 (Ishida Y, et al. EMBO J 11: 3887-3895, 1992) and later shown to be a negative regulator of the immune system (Nishimura H, et al. Immunity 11: 141-151, 1999), and in 2002, an antibody against PD-L1, a ligand of PD-1, showed therapeutic effect on cancers in a mouse model (Iwai, Y. et al. Proc Natl Acad Sci U.S.A 99: 12293-12297, 2002). As a clinical application of PD-1 antibody, nivolumab was showed to be effective against unresectable malignant melanoma (Topalian SL, et al. N Engl J Med 366: 2443-2454, 2012). After long development processes, ICIs have recently been widely used for the treatment of melanoma, non-small cell lung cancer, renal cancer, and hematologic malignancy, not only for metastatic cases but also in the adjuvant setting. According your comment, we have added the sentences associated with history research on ICIs in section 6.1.

  1. This reviewer personally misses some important findings regarding predictive and prognostic factors in HCC patients treated with sorafenib. Indeed, unlike other targeted therapies, predictive and prognostic markers in HCC patients treated with sorafenib are lacking. Their identification could help clinicians in the daily management of these patients, mostly in light of the new therapeutic options available in the first (please refer to PMID: 31640191).

Response: Thank you very much for your comments. We also agree with your opinion.

 As the variety of treatment options increases, the evidence for markers that predict the therapeutic efficacy of MTAs and prognosis of HCC patients treated with MTAs, including sorafenib, becomes more important. This information can help clinicans choose from new therapeutic options available (Brunetti O, et al. Medicina 55(10), 707, 2019). There is some evidence of biological prognostic markers for sorafenib treatment that has been most clinically used in advanced HCC. Angiogenetic markers, including angiopoietin-2 (Ang-2) and VEGF, have been analyzed in several studies. In the SHARP study, baseline VEGE and Ang-2 plasma levels could not be used as prognostic biomarkers for HCC patients treated with sorafenib and placebo (Llovet J, et al. M Engl J Med 359(4): 378-390, 2008). In contrast, it was reported that decreases in plasma VEGF levels due to sorafenib treatment after 8 weeks was a predictor of better median OS (Tsuchiya K, et al. Cancer 120(2): 229-237, 2014). Another study showed that HCC patients who were refractory to sorafenib treatment had higher circulating cell-free DNA (cfDNA) levels of VEGF and that the group with cf DNA levels of VEGF above median had worse time to progression (TTP) (Oh C, et al. BNC Cancer 19(1): 292, 2019). A study evaluating SNPs of multiple angiogenic molecules including HIF-1α, VEGF, and Ang-2, demonstrated that rs12434438 of HIF-1α, rs2010963 of VEGF-A, and rs4604006 of VEGF-C were dependent factors predicting the therapeutic effects of sorafenib (Faloppi L, et al. Target Oncol 15(1): 115-126, 2020). In the inflammatory TME, circulating immune cells and cytokines also play important roles in HCC progression and could be prognostic markers for sorafenib treatment. A high neutrophil- lymphocyte ratio (NRL) has been reported to be a useful biomarker for predicting poor therapeutic effect of sorafenib (Lue A, et al. Oncotarget 8(61): 103077-103086, 2017), and a high NLR before sorafenib treatment is reported to be associated with short OS (Zheng J, et al. Cell Physiol Biochem 44(3): 967-981, 2017). Insulin-like growth factor 1 (IGF-1) also has the potential to predict the therapeutic effect of MTAs, including sorafenib. It has been reported that HCC patients with high baseline IGF-1 achieve higher disease control rate (DCR) and show longer PFS and OS (Shao YY, et al. Clin Cancer Res 18(14): 3992-3997, 2012).

 According your comment, we have added the sentences associated with predictive and prognostic markers in HCC patients treated with sorafenib in section 5.

  1. The authors need to come up with a table to show the role of TME cells as Checkpoint for Immunological Patrolling in HCC to highlight their exact role.

Response: Thank you very much for your comments. We also agree with your opinion.

According your comment, we have added the table to show the role of TME cells in section 4.

  1. The authors need to highlight what new information the review is providing to enhance the research in progress.

Response: Combination therapies including ICIs may central roles in HCC treatment, and there are several clinical trials currently ongoing in combination therapies such as Pembrolizumab plus lenvatinib treatment, durvalumab (PD-L1 antibody) and tremelimumab, and nivolumab and ipilimumab (CTLA-4 antibody). The results of phase III trials are awaited, and the overviews of the results in previous clinical trials are provided as latest knowledge in our review. We also describe the results of pre-clinical studies using immunological classification and animal models, which will contribute to development of biomarkers that the efficacy of treatment and patient prognosis.

 In a phase 1b trial with pembrolizumab plus lenvatinib treatment for HCC patients with Child-Pugh A, favorable results were achieved, with a median PS of 22.0 months, PFS of 9.3 months, and ORR per modified Response Evaluation Criteria in Solid Tumors (mRESIST) of 46.0% (Finn RS, et al. J Clin Oncol 38(26): 2960-2970, 2020). A phase III trial (LEAP 002) comparing lenvatinib plus pembrolizumab combination therapy with lenvatinib monotherapy is currently ongoing, and additional evidence of efficacy of combination therapy is expected.

 In 2020 at ASCO, combination therapy with durvalumab (PD-L1 antibody) and tremelimumab showed promising outcomes, with a median OS of 18.7 months, median PFS of 2.2 months, and ORR per RESIST v1.1 of 24.0% (Kudo M, et al. Liver cancer 8(4): 221-238, 2019). Combination therapy with durvalumab and tremelimumab is likely promising, and the phase III trial (HIMALAYA) is currently ongoing.

 Combination therapy with nivolumab and ipilimumab (CTLA-4 antibody) is reportedly effective in patients with advanced HCC. In a phase I/II trial (CheckMate 040), this combination therapy showed a median OS of 22.8 months, and ORR per RESIST v1.1 was 32% (Yau T, et al. JAMA Oncol 6(11): e204564, 2020). A phase III trial (CheckMate 9DW) comparing the combination of nivolumab and ipilimumab with lenvatinib or sorafenib is currently ongoing.

 The positive results of the IMBrave trial have positioned combination atezo+bev as the first-line treatment for advanced HCC, but subsequent sequential MTA therapy is still important. It has been reported that the binding of nivolumab to CD8+ T cell lasts for more than 20 months in the treatment for lung cancer (Osa a, et al. JCI Insight 3(19): e59125, 2018). Even case where PD-1/PD-L1 antibody leads to progressive disease, post-treatment with MTAs change TME from immune suppressive to immune permissive. Sequential ICI-MTA therapy is expected to bring about TME changes similar to those from PD-1/PD-L1 antibody plus MTA combination therapy and to show a favorable therapeutic effect (Kudo M, et al. Cncers 12(5): 1089, 2020) (Joerger, M, et al. J Gastrointest Oncol 10(2): 373-378, 2019)

 Furthermore, it is interesting that there are increasing studies on the prediction of therapeutic effect of ICIs. Recent studies indicate that ICIs have no effect on patients with WNT/β-catenin-mutated HCC, an immune exclusion subclass (Llovet JM, et al. Nat Rev Clin Oncol 15(10): 599-616, 2018). It has also reported that the effectiveness of MTAs such as sorafenib is not related to the WNT/β-catenin mutation status (Harding JJ, et al. Clin Cncer Res 25(7): 2116-2126, 2019). Although the frequency of WNT/β-catenin-mutated HCC, 20-30%, is not high (Pinyol R, et al. Clin Cncer Res 25(7): 2021-2021, 2019), MTAs rather than ICIs may be suitable as first-line treatment for WNT/β-catenin-mutated HCC. Furthermore, WNT/β-catenin-mutated HCC show an iso-high intensity in the hepatobiliary phase of gadolinium ethoxybenzyl diethylenetriaminepentaacetic acid-enhanced magnetic resonance imaging (Gd-EOB-DTPA-MRI: EOB-MRI) (Kitao A, et al. Radiology 275(3): 708-717, 2015). EOB-MRI may be useful for predicting the state of WNT/β-catenin mutation in HCC and treatment resistance to ICIs

 According your comment, we have added the sentences in section 6.1. and 6.2.

Reviewer 2 Report

Kyoko Oura and colleagues have written a well-researched review of the literature of hepatocellular carcinoma.

Minor comment, line 485 should be modified to indicate that FOXP3 is a transcription factor whose expression is restricted primarily to the nucleus, and is not expressed on the surface of T cells. 

Author Response

  1. Minor comment, line 485 should be modified to indicate that FOXP3 is a transcription factor whose expression is restricted primarily to the nucleus, and is not expressed on the surface of T cells.

Response: Thank you very much for your comments. We also agree with your opinion.

The surface of tumor-infiltrating lymphocytes (TILs) contains several antigens, including CD3, CD4, CD8, CD16, CD20, CD56, CD57, CD68, and CD169 (Ding W, et al. Medicine 97(50): e13301, 2018). FOXP3 is the master transcription factor localized in the nucleus of Tregs. FOXP3+ Treg are representative TILs that function as tumor suppressors. In this manuscript, we discuss the possibility that TILs containing FOXP3+ T cells are biomarkers associated with the prognosis of HCC patient. Although there is a report that high filtration of FOXP3+ T cells is associated with poor prognosis (Katz SC, et al. Ann Surg Oncol 20(3): 946-955, 2013), others have reported no correlation between cell infiltration and OS (Wang F, et al. Liver Int 32(4): 644-655, 2012). A high density of CD8+ T cells has been shown to have a poor prognosis [121], but another report indicated the opposite result (Ramzan M, et al. Liver Int 36(3): 434-444, 2016). The impact of TILs on the prognosis of HCC is controversial, but a recent meta-analysis has shown that the density of FOXP3+, CD8+, CD3+, and granzyme B+ lymphocytes in TILs is significantly associated with improved survival, and these subsets can serve as prognostic biomarkers in HCC (Ding W, et al. Medicine 97(50): e13301, 2018).

According to the comment, we have altered the corresponding part as mentioned above.

Round 2

Reviewer 1 Report

Unfortunately, the authors described the side effects of the therapy very generally. The characteristics of the side effects of the new therapies are important for safety. Therefore, I believe that the authors could have described this issue in more detail.

Specifically: 

Immune Checkpoint Inhibitor-Related Myositis represents a novel frontier of immune-oncology and Immune checkpoint inhibitor (ICI)-related inflammatory diseases, including polymyositis (PM) and dermatomyositis (DM), in patients suffering from neoplastic disorders represent a medical challenge. The treatment of these conditions has taken on new urgency due to the successful and broad development of cancer-directed immunological-based therapeutic strategies. While primary and secondary PM/DM phenotypes have been pathophysiologically characterized, a rational, stepwise approach to the treatment of patients with ICI-related disease is lacking. In the absence of high-quality evidence to guide clinical judgment, the available data must be critically assessed. Partially neglected immunological and clinical findings have been recently reviewed to obtain insights into the biological profiles of ICI-related PM/DM and potential treatment options. Differential diagnosis is therefore essential to stratifying patients according to prognosis and therapeutic impact.

Moreover, rare but clinically significant ICI-related adverse events (IrAEs) such as lymphocytosis associated with fatal hepatitis have also been reported; indeed, recent advances in tumor immunotherapy have made it possible to efficiently unleash immune effectors, reacting against neoplastic cells. Although these approaches primarily aim to eradicate malignancy, immune-related adverse events often influence patients' prognosis, constituting a new spectrum of side effects. Taking into account the typical microenvironment and the intricate equilibrium between the anti-tumor response and the immune cells, the HCC constitutes a unicum in the immune-oncology field. The clinical proficiency of the immune checkpoint inhibitors in cancer patients warrants timely prevention and management of off-target consequences in order to optimize this promising therapeutic option. A comprehensive but synoptic assessment of druggable targets and a stepwise patient-oriented approach for the treatment of ICI-related PM/DM, common and rare IrAEs would be useful

There are also some typos (i.e. ipilimab instead of ipilimumab). Please carefully check on them.

Author Response

  1. Unfortunately, the authors described the side effects of the therapy very generally. The characteristics of the side effects of the new therapies are important for safety. Therefore, I believe that the authors could have described this issue in more detail.

Response: Thank you very much for your comments. We agree with your opinion.

 While ICIs inhibit negative feedback mechanisms of the immune system resulting very durable anti-tumor responses through blockade of CTLA-4 or PD-1, these immune-related adverse events (irAEs) can interfere with the immunosuppressive therapy. ICIs can influence peripheral tolerance to autoantigens, resulting in antibody formation, which causes damage to various organs. ICIs also release T cells, with subsequent production of pro-inflammatory cytokines such as IFN-γ and TNF, resulting in excessive autoimmunity tumor inflammation. The onset of irAEs is unpredictable because patients can emerge irAEs early after ICI treatment up to more than 18 months after initiation of treatment (Khan S, et al. Semin Cancer Biol 64: 93-101, 2020), and may develop multiple different irAEs either simultaneously or subsequently (Wang D, et al. JAMA Oncol 4(12): 1721-1728, 2018). The frequency of irAE has been reported to be 60%-85% in patients treated with CTLA-4 antibody, 57%-85% in patients treated with anti-PD-1, and 95% in patients treated with the combination of anti-CTLA-4 and anti-PD-1(Haanen J, et al. Ann Oncol 28: 119-142, 2017). While most irAEs are reversible and can be addressed, it is important to recognize fatal irAEs. According to meta-analysis of anti-CTLA-4 or anti-PD-L1 monotherapy, common target organs with irAEs were the skin, gastrointestinal tract, endocrine glands, and the liver (Bertrand A, et al. BMC Med 13: 211, 2015). The most frequent fatal irAEs were pneumonitis, hepatitis, and neurotoxic effects. Combination with anti-CTLA-4 and anti-PD-(L)1 is prone to appear incrementally toxicity. Patients treated with anti-CTLA-4 and anti-PD-(L)1 combination had a higher risk and more frequent multi-organ involvement than ICI monotherapy (Zhang B, et al. Int Immunopharmacol 63: 292-298, 2018). The incidence rates of Grade 3 or higher in the Common Terminology of Clinical Adverse Events (CTCAE) categorization was 41%, and the common fatal events were myocarditis, myositis, and neurologic events. Although fatal irAEs remains rare, irAEs should be recognized promptly as early interventions may alleviate future complication.

 In general, the incidence of irAEs in HCC patients treated with ICIs including nivolumab, pembrolizumab, and ipilimumab is not significantly different from other cancer types. The incidence of severe irAEs tended to be lower in HCC, ranging from 10%-20%, compared to more than 30% in other cancers (Sangro B, et al. J Hepatol 72(2): 320-341, 2020). The incidence of hepatic irAEs is higher in HCC patients, possibly due to the unique liver immunobiology and chronic inflammatory conditions of the liver such as cirrhosis and viral hepatitis. While only 3% of patients experienced hepatitis enzymes elevation and 0.85% of patients experienced treatment-related hepatitis among other cancer types (Wang Y, et al. JAMA Oncol 5(7): 1008-1019, 2019), 13.7% of HCC patients treated with anti-PD-1 monotherapy experienced hepatitis enzymes elevation of any grade (Voutsadakis IA, et al. Hepatobiliary Pancreat Dis Int 18(6): 505-510, 2019). Cutaneous toxicity is the first common irAE among patients treated with ICIs, the incidence of rash and pruritus in HCC patients treated with nivolumab and pembrolizumab monotherapy was 11-23% and 10-11.5%, and 13-19% and 12-18.3%, respectively (Cui T, et al. Onco Targets Ther 13: 11725-11740, 2020). As digestive system irAEs, diarrhea is also a frequent adverse reaction due to ICIs. Diarrhea occurred in 10% of HCC patients treated with nivolumab, 11% treated with pembrolizumab, and 12-24% treated with nivolumab and ipilimumab combination. Colitis is a serious irAE that can cause fatal colonic perforation and peritonitis. Most ICI-induced colitis in HCC patients is grade 3 and occurs in 1% of those treated with PD-L1 antibody and 2.6% of those treated with CTLA-4 antibody and PD-L1 antibody combination. The incidence of irAEs among HCC patients treated with ICIs was relatively low in the other targeted organs. The incidence of pneumonia in HCC patients treated with nivolumab was approximately 3%, and there are few patients with grade 3 or higher. In phase II pembrolizumab clinical trial, 7% of HCC patients reported myalgia, but no patients were found in phase III trial. In other ICIs trials for HCC, the incidence of heart-related complication was extremely rare, less than 1%. In clinical trials of tremelimumab for HCC, 15% of those patients developed grade 3 or higher encephalopathy, yet these complications are more associated with underlying cirrhosis than ICI treatment. The incidence of ICI-induced myositis was also rare, and the frequency during ICI treatment for HCC is unknown, but it is a cautionary irAE that can develop suddenly and progress rapidly (Kadota H, et al. Curr Rheumatol Rep 21(4): 10, 2019). Despite the low incidence, if left untreated, complications related to the respiratory, circulatory, and nervous systems can be fatal. For rare irAE, early diagnosis and proper management are important to improving the outcome of ICI treatment.

 As VEGF targeted therapies can facilitate drug delivery to HCC tissues, combination with ICIs and VEGF inhibitor may reduce the dose of ICIs, thus reducing the toxicity of ICI compared to ICI monotherapy (Fukuhara D, et al. Nat Rev Clin Oncol 15(5): 325-340, 2018). According to phase III clinical trial of atezo/bev combination therapy for HCC, the incidence of all grades and high grade irAE in patients treated with atezo/bev combination was not significantly different from anti-PD-1 monotherapy or PD-1 antibody plus CTLA-4 antibody (Finn R, et al. N Engl J Med 382(20): 1894-1905, 2020). According to the assessment of the common irAEs in ICI monotherapy, patients treated atezo/bev combination had lower incidence of cutaneous irAEs with 12% rash and 19.5% pruritus. The incidence diarrhea was not significant difference, and less than 10% of patients reported abnormal thyroid function. The most common AE in atezo/bev combination therapy was hypertension with the incidence of 29.8%. 56.5% of patients treated with this combination had some Grade 3 or 4 side effects compared with 55.1% of patients were treated with sorafenib. The side effects of Grade 3 or higher were hypertensive encephalopathy (15.2%), bleeding (7.9%), myelosuppression (8.5%), infection (8.2%), and nephrotic syndrome (3.0%), and most are influenced by bevacizumab. As the use of ICIs continues to expand, early detection and management of irAEs are crucial to maximize the duration of treatment while minimizing toxicities for patients. The safety of ICIs in the setting of advanced liver cirrhosis and their efficacy in different etiologies remain open questions and need to be addressed in future trials.

 According to your comment, we have provided a new section associated with irAEs.

  1. There are also some typos (i.e. ipilimab instead of ipilimumab). Please carefully check on them.

Response: Thank you very much for your comments. According to the comment, we have corrected the error.

Round 3

Reviewer 1 Report

The authors addressed all my concerns by answering appropriately.